# Itaconate transport across the plasma membrane and *Salmonella*-containing vacuole via MCT1/4 modulates macrophage antibacterial activity

Qingcai Meng [1,3], Chengxi Li[1,3], Yuping Cai [2], Ying Chen[1], Xiaoqing Chen[1], Xin Wang[1], Biling Zhang[1], Yue Zhang[1], Feng Liu[1] & Meixin Chen [1] ✉

Itaconate accumulates in macrophages upon bacterial infection, and manifests antibacterial activity. Convincing evidence substantiates that itaconate is transported across the plasma membrane and vacuolar membrane, but the molecular bases underlying bidirectional transport of itaconate across membranes and its effects on intracellular bacterial replication are less known. Here, we identify MCT1 and MCT4 as bidirectional transporters of itaconate. In addition to modulating itaconate concentration as transporters at the plasma membrane, MCT1 and MCT4 function as itaconate transporters at *Salmonella*-containing vacuole (SCV). Upon *Salmonella* infection, MCT1 and MCT4 transport itaconate into SCV facilitated by RAB32. Itaconate is also secreted out of cells through MCT1 and MCT4 as the infection persists. The suppression of MCT1 and MCT4-dependent itaconate secretion increases the overall concentration of itaconate and the proportion of itaconate-targeted *Salmonella* intracellularly, consequently inhibiting *Salmonella* replication. Our study thus offers valuable insights into itaconate transport during bacterial infection and provides proof of principle for the development of itaconate-dependent therapeutic strategies.

The bacterial infection leads to a dramatic metabolic shift in cells of hematopoietic origin, which is a crucial event of cell-autonomous defenses[1–3]. The activation of macrophages by bacteria undergoes enhanced glycolysis and disturbed tricarboxylic acid (TCA) cycle. The break at isocitrate dehydrogenase 1 (IDH1) of the TCA cycle results in increasing cis-aconitate[4]. Subsequently, the highly expressed immune-responsive gene 1 (IRG1, also known as aconitate decarboxylase 1, ACOD1) catalyzes the decarboxylation of cis-aconitate into itaconate[2,5]. Itaconate has been reported to regulate immune responses by modifying proteins known as "itaconation"[6] as well as "itaconylation"[7,8]. Alternatively, itaconate works as a well-established antibacterial metabolite[9–12], coordinated decently by the RAB32-LRRK2 complex as illuminated in our previous study[13,14]. In the context of antibacterial immunity and immunoregulation, maintaining optimal intracellular itaconate concentration is crucial[15,16]. And the potential transport of itaconate through plasma membrane or itaconate delivery into bacteria-containing vacuole has been mentioned in various scenarios[13,17–20]. The bidirectional transporters for itaconate likely exist, given its limited cell permeability[16,21]. More recently, ABCG2 has been recognized as an exporter of itaconate at plasma membrane[22]. But the absence of ABCG2 did not completely block the secretion of itaconate from macrophages[22], which prompts us to study more about itaconate

[1]Institute of Infectious Diseases, Shenzhen Bay Laboratory, Shenzhen, China. [2]Interdisciplinary Research Center on Biology and Chemistry, Shanghai Institute of Organic Chemistry, Chinese Academy of Sciences, Shanghai, China. [3]These authors contributed equally: Qingcai Meng, Chengxi Li. ✉e-mail: chenmx@szbl.ac.cn

transport. Overall, currently, the bidirectional transporter(s) at the plasma membrane or bacteria-containing vacuole remain unknown[21].

Solute carrier (SLC) proteins belong to a major transporter superfamily[23], which has been demonstrated as metabolic gatekeepers at the plasma membrane or vacuoles in various cell types, including macrophages[24,25]. For instance, MCT1 (SLC16A1) facilitates succinate release from muscle tissue[26]. SLC13A5 is a well-established citrate transporter[27]. SLC7A11 primarily functions as a cystine/glutamate exchanger, focusing on transporting cystine and glutamate[28]. SLC1A5 primarily transports neutral amino acids, especially glutamine[29]. SLC39A7 facilitates the uptake of zinc ions into cells, playing a critical role in zinc homeostasis. SLC22A1, also known as OCT1 (Organic Cation Transporter 1), is involved in the transport of organic cations[30]. However, little is known about the role of SLC proteins in transporting antibacterial metabolites during bacterial infection.

In this study, we identify MCT1 (SLC16A1) and MCT4 (SLC16A3) as key bidirectional itaconate transporters localized at both the plasma membrane and the bacteria-containing vacuole (BCV), where they regulate itaconate trafficking across cellular membranes. We demonstrate that acidic pH facilitates itaconate transport by promoting its conversion to the monocarboxylic form. Furthermore, we show that RAB32 directs MCT1/4 localization to the Salmonella-containing vacuole (SCV), thereby enhancing itaconate import into this compartment. Our findings connect the metabolic transporters MCT1/4 and anti-bacterial defense by dissecting their important roles in controlling intracellular itaconate distribution.

## Results

### The inhibition of MCT1 and MCT4 transport activity resists Salmonella infection

We first verified itaconate secretion in bacteria-infected macrophages. Itaconate accumulated in the cytosol (-0.2–3 nmol/10⁶ RAW264.7 cells) and was subsequently released into the cell supernatant (-2–30 nmol/10⁶ RAW264.7 cells) over time following infection with the gram-negative bacteria Salmonella (Fig. 1a, b). A larger proportion of itaconate was secreted into the extracellular space during infection (Fig.1c), analogous to that observed in LPS-treated macrophages[20,31]. Based on this point, we reasoned that if itaconate transport across the plasma membrane has been dampened, the concentration of intracellular itaconate would be altered pronouncedly. As a result, it may influence the bacteria-eliminating effects of itaconate. Thus, to search for itaconate transporter(s), we first determined whether some available inhibitors of SLC transporters modulated the bacterial survival in cells. The concentrations of SLC inhibitors (for MCT1, MCT4, SLC13A5, SLC7A11, etc.) were established based on documented methodologies[28,29,32–40]. The survival of Salmonella enterica serovar Typhi (Salmonella Typhi) was remarkably inhibited in the presence of Su3118 (Syrosingopine, a dual inhibitor for MCT1 and MCT4)[41] in RAW264.7 cells (Fig. 1d, e) and bone marrow-derived dendritic cells (Supplementary Fig. 1a). We further found that Su3118 attenuated the replication of the broad-host-range serotype Salmonella enterica serovar Typhimurium (Salmonella Typhimurium) (Fig. 1f). VB124[42] (a MCT4 inhibitor) in combination with AZD3965[43] (a MCT1 inhibitor) also had a strong inhibitory effect on Salmonella proliferation in RAW264.7 cells (Fig. 1g). Importantly, a lower bacterial load was detected in human monocyte-derived macrophages (hMDM) with MCT1 and MCT4 impairments (Fig. 1h). And these inhibitors themselves had no significant influence on Salmonella growth in a mammalian-cell-free system (Supplementary Fig. 1b–d). We also examined whether MCT1 or MCT4 (MCT1/4) inhibitors inhibited the phagocytic capacity of cells, which probably led to the detection of a lower bacterial load in cells. We found that long-term inhibition of MCT1 and MCT4 transport activity did not alter the entry of Salmonella into bone marrow-derived macrophages (BMDM), RAW264.7 cells, or DC2.4 cells (Supplementary Fig. 1e–h). It indicates that blocking MCT1/

4 does not affect macrophage phagocytosis. We then constructed MCT1/4 depletion cell line by CRISPR/Cas9 (Supplementary Fig. 1i). It is worth noting that Mct4⁻/⁻ RAW264.7 cells treated with the MCT1 inhibitor (AZD3965) sustained less bacterial replication (Fig. 1i). And the MCT4 inhibitor VB124 dramatically reduced bacterial load in Mct1⁻/⁻ RAW264.7 cells (Fig. 1j). These data highlighted that the combinational defect in MCT1 and MCT4 enhances the antibacterial activity in macrophages.

However, the bactericidal effect of Su3118/(AZD3965 + VB124) disappeared or was reduced in Irg1-deficient RAW264.7 cells or HeLa cells, which lack endogenous itaconate generation (Fig. 1k, l and Supplementary Fig. 1j, k). IRG1 expression in HeLa cells rescued the bacterial-killing properties of Su3118/(AZD3965 + VB124) (Fig. 1m). Consistently, the ablation of Irg1 in mice led to the inefficiency of Su3118 in reducing Salmonella load (Fig. 1n). Besides, we observed that the protein level of IRG1 and MCT4 concurrently increased in RAW264.7 cells, spleen, and liver after bacterial infection with a modest change in mRNA level of MCT1/4 (Fig. 1o, p and Supplementary Fig. 1l–o). Taken together, these data underlined the critical roles of MCT1 and MCT4 transport activity in itaconate-dependent immune responses against bacterial infection, which predicted that the solute carriers MCT1 and MCT4 were itaconate transporters.

### The inhibition of MCT1 and MCT4 transport activity promotes the prevalence of itaconate-targeted Salmonella

We next continued to explore the roles of MCT1 and MCT4 in regulating itaconate-based bacterial killing. Markedly, MCT1 and MCT4 were detected to be located at the plasma membrane in macrophages (Fig. 2a and Supplementary Fig. 2a, b). The transport activity of MCT1 and MCT4 at the plasma membrane may modify the level of itaconate within cells. To better understand whether the MCT1 and MCT4-related restriction of bacteria growth is metabolites-related, we profiled the metabolic changes of BMDMs under various conditions. We have observed an accumulation of itaconate in BMDMs with disrupted MCT1 and MCT4 transport activity through untargeted metabolomics (Fig. 2b). Spontaneously, succinate, lactate and pyruvate, which are also substrates of MCT1 or MCT4, accumulated (Fig. 2b). The inhibition of MCT1 and MCT4 also increased the concentration of itaconate in RAW264.7 cells at the late stage of infection (Fig. 2c). We also scrutinized whether a higher concentration of itaconate resulted in a greater number of itaconate-targeted Salmonella Typhi. To do that, we took advantage of the nanoluciferase-biosensor of itaconate located in bacteria (Supplementary Fig. 2c)[13]. We found that more bacteria were targeted by itaconate in RAW264.7 cells when MCT1 and MCT4 activity was suppressed at the later phase of infection (Fig. 2d and Supplementary Fig. 2d, e). Importantly, we also found that more Salmonella were targeted by itaconate in hMDM with the treatment of Su3118, detected by using itaconate biosensor (Fig. 2e). Furthermore, administration of Su3118 in WT mice or administration of AZD3965 in Mct4⁻/⁻ mice significantly promoted the accumulation of intracellular itaconate in the spleen, leading to the detection of higher levels of itaconate by itaconate biosensor in Salmonella (Fig. 2f, g). As a result, a lower bacterial load was detected in spleen from mice with MCT1 and MCT4 impairments shown in Fig.1n. However, in macrophages infected with Salmonella Typhi for a shorter period (-7 h), Su3118 failed to modulate the level of the intracellular itaconate (Fig. 2h). This may explain why the percentage of itaconate-targeted Salmonella Typhi at around 7 h changed modestly in RAW264.7 cells with MCT1 and MCT4 deficiency (Fig. 2i). In consequence, the replication of Salmonella in RAW264.7 cells with the weak activity of MCT1 and MCT4 was strongly inhibited at late but not early (-7 h) time-point of infection (Fig. 1 and Supplementary Fig. 2f, g). Considering a lower proportion of itaconate was secreted at 7–8 h post-infection (Fig. 1c), it is predictable that the effect of the inhibitors on itaconate concentration or Salmonella killing in the initial phase of infection is rather limited. With the extension of

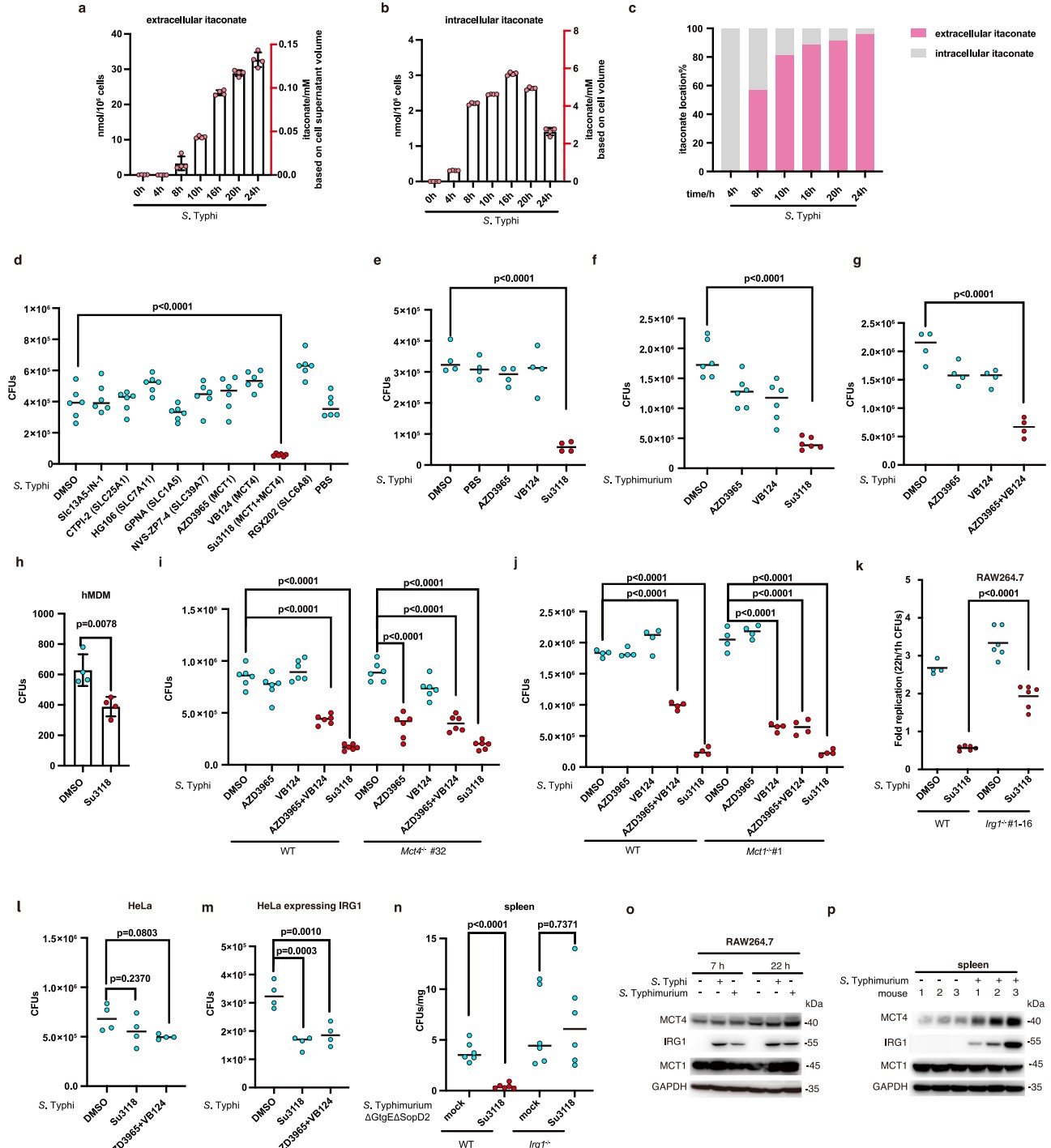

**Fig. 1 | The inhibition of MCT1 and MCT4 transport activity resists *Salmonella* infection.** Extracellular (**a**) and intracellular (**b**) itaconate in *Salmonella* (*S.*) Typhi-infected RAW264.7 cells were measured by LC-MS, with the intracellular/extracellular percentage shown in (**c**). **d** RAW264.7 cells treated with indicated inhibitors were infected with *S.* Typhi, and bacteria were counted at 22 h. Su3118 (10 μM), AZD3965 (1 μM), VB124 (20 μM), RGX-202 (5 μM), GPNA (20 μM), NVS-ZP7-4 (20 nM), SLC13A5-IN-1 (0.044 μM), HG106 (5 μM), CTPI-2 (3.5 μM). RAW264.7 cells treated with the indicated inhibitors (Su3118 10 μM, AZD3965 1 μM, VB124 20 μM) were infected with *S.* Typhi (**e, g**) or *S.* Typhimurium (**f**) for 22 h, and bacteria were counted. **h** Human monocyte-derived macrophages (hMDM) treated with DMSO/Su3118 (5 μM) were infected with *S.* Typhi for 22 h, and bacteria was counted. **i, j** WT (wild-type), *Mct4⁻/⁻* or *Mct1⁻/⁻* RAW264.7 cells treated with the indicated inhibitors were infected with *S.* Typhi for 22 h, and intracellular bacteria was examined. WT or

*Irg1⁻/⁻* RAW264.7 cells (**k**), HeLa cells (**l**) or HeLa expressing IRG1 (**m**) treated with the indicated inhibitors were infected with *S.* Typhi for 22 h, and bacteria were counted. **n** C57BL/6 (WT) or *Irg1⁻/⁻* mice were intraperitoneally (I.P.) injected Su3118 (5 mg/kg). 24 h later, mice were infected I.P. with *S.* Typhimurium ΔGtgEΔSopD2 (10⁴ CFU) for 24 h. Bacterial loads were analyzed. RAW264.7 cells (**o**) were infected with *Salmonella* for 7 or 22 h. The spleen lysate (**p**) was collected from mice injected ± Su3118 (5 mg/kg) and I.P. infected with *S.* Typhimurium (10² CFU). Lysates were analyzed by immunoblotting with anti-MCT1, MCT4, IRG1, or GAPDH antibodies. Data are shown as the mean ± SD. Independent biological replicates-*n* = 4 (**a, b, e, g, h, j, l, m**); *n* = 6 (**d, f, i, k, n**). A two-tailed Student's *t*-test was conducted for pairwise comparisons (**h, n**), while one-way ANOVA was used for multiple comparisons involving a single independent variable (**d–g, i–m**). Source data are provided as a Source data file.

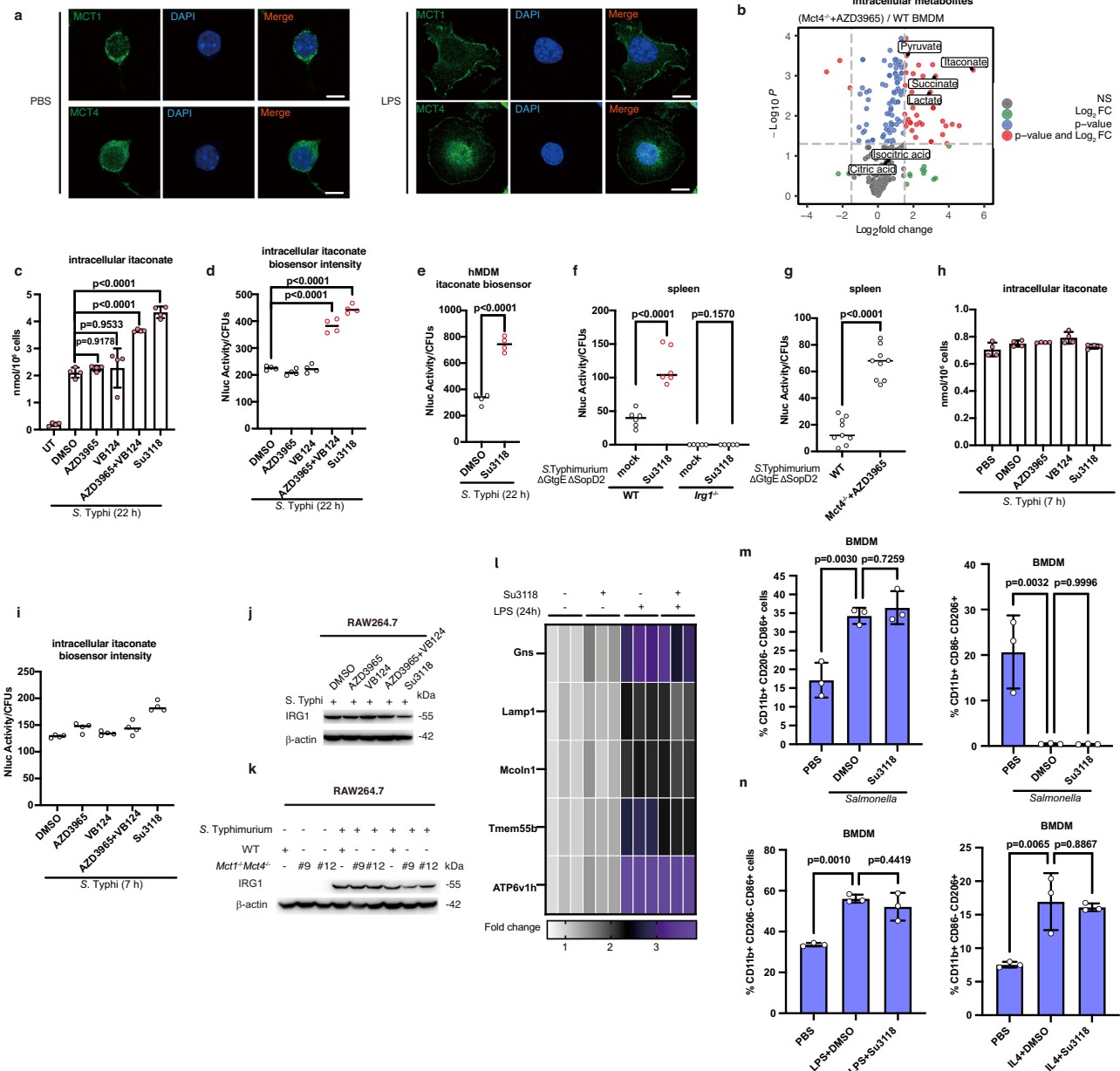

**Fig. 2 | The inhibition of MCT1 and MCT4 transport activity promotes the proportion of itaconate-targeted *Salmonella*. a** RAW264.7 cells ± LPS (100 ng/ml, 24 h) were stained with MCT1 or MCT4 (green) antibodies and DAPI (blue), and imaged. Scale bar, 10 μm. **b** Volcano plot of non-targeted metabolomics in AZD3965-treated *Mct4*⁻/⁻ and WT BMDM ± LPS (100 ng/ml; AZD3965 0 or 100 nM). Red dots indicate significantly changed metabolites ($p < 0.05$, |Log2 FC| > 1.5; $n = 6$ biologically independent samples per group). FDR-adjusted p-values by two-tailed Student's t-test. **c, d, h, i** Intracellular itaconate from RAW264.7 treated with the indicated inhibitors (Su3118 10 μM, AZD3965 1 μM, VB124 20 μM) and infected by *S*. Typhi was determined by LC-MS or itaconate biosensor. **e** Itaconate from hMDM treated with Su3118 (5 μM), and infected by *S*. Typhi was determined by itaconate biosensor. **f, g** WT, *Irg1*⁻/⁻, or *Mct4*⁻/⁻ mice were injected I.P. with Su3118 (5 mg/kg) or AZD3965 (100 mg/kg), infected I.P. with *S*. Typhimurium ΔGtgEΔSopD2 (10⁴ CFU) for 24 h, and

spleen nanoluciferase was measured. **j, k** RAW264.7 cell **j** treated with the inhibitors (Su3118 10 μM, AZD3965 1 μM, VB124 20 μM) were infected with *S*. Typhi for 22 h. WT or *Mct1*⁻/⁻*Mct4*⁻/⁻ RAW264.7 cells (**k**) were infected with *S*. Typhimurium for 22 h. Lysates were immunoblotted for IRG1 and β-actin. **l** Relative mRNA levels of genes in BMDM treated with Su3118 (5 μM, 24 h), ± LPS (10 ng/ml, 24 h), were measured by RT-PCR, normalized to untreated cells, and shown as heatmaps (triplicates). **m, n** Flow cytometry of BMDM (infected with *Salmonella* or treated with Su3118, LPS, or IL-4) showing percentages of CD11b⁺CD86⁺CD206⁻ and CD11b⁺CD86⁻CD206⁺ cells. A two-tailed Student's t-test was conducted for pairwise comparisons (**e–g**), while one-way ANOVA was used for multiple comparisons involving a single independent variable (**c, d, m, n**). Data are shown as the mean ± SD. Independent biological replicates-$n = 3$ (**l, m, n**); $n = 4$ (**c–e, h, i**); $n = 6$ in WT and $n = 5$ in *Irg1*⁻/⁻ mice in **f** $n = 9$ (**g**). Source data are provided as a Source data file.

infection, the proportion of secreted itaconate is larger (Fig. 1c). So, in the later period, MCT1 and MCT4 inhibitors effectively control the secretion of itaconate, leading to increased cellular itaconate and also the attenuated growth of bacteria. To see the influence of MCT1 and MCT4 deficiency on the protein level of IRG1, we analyzed whether the protein level of IRG1 was upregulated by the inhibitors. The results have shown that these inhibitors did not change IRG1 protein level

(Fig. 2j and Supplementary Fig. 2h). The double deficiency of MCT1 and MCT4 also did not raise the IRG1 protein level (Fig. 2k).

Given itaconate also has a role in regulating lysosomal biogenesis[8], we measured that whether blocking MCT1 and MCT4 affected lysosomal gene expression. LPS activated the expression of lysosomal genes (Supplementary Fig. 2i). Su3118, the MCT1 and MCT4 inhibitor, hardly regulated the expression of the lysosomal genes

(Fig. 2l). Su3118 also did not affected the protein level of Lamp1 (Supplementary Fig. 2j). Overall, blocking MCT1/4 probably did not boost the expression of lysosomal genes.

It is possible that macrophage polarization participate in itaconate-based antibacterial effect[8]. Blocking MCT1 and MCT4 had minimal impact on the polarization of macrophages under during bacterial infection (Fig. 2m). Representative flow plot from BMDMs were shown (Supplementary Fig. 2k). Furthermore, the polarization of the majority of macrophages under LPS/IL4 stimulation were not influenced by the inhibition of MCT1 and MCT4 (Fig. 2n). Su3118 also did not significantly change the mRNA level of *Cd206* or *Cd86* in macrophages in response to bacterial infection or LPS/IL4 treatment (Supplementary Fig. 2l). These data suggest that the inhibition of MCT1/4 does not change M1/M2 polarization of macrophages.

We next also ruled out that the antibacterial immunity was attributed to the effect of lactate, pyruvate, or succinate, which were no reported to be transported by MCT1/4. These metabolites showed no antibacterial activity (Supplementary Fig. 3a–c) as documented[44–46]. Therefore, our findings indicate that the inhibition of MCT1 and MCT4 increases the general amount of itaconate intracellularly, and naturally enhances the antibacterial efficacy of itaconate by promoting the prevalence of itaconate-targeted *Salmonella*.

### MCT1 and MCT4 transport intracellular itaconate out of cells

To see whether MCT1 and MCT4 transport itaconate out of cells, we have developed an efficient assay to examine itaconate secretion (Fig. 3a and Supplementary Fig. 2c), based on an itaconate biosensor as delineated[13]. We have proved that the eGFP-biosensor of itaconate showed competence in detecting itaconate secretion of LPS-activated RAW264.7 cells (Supplementary Fig. 4a, b), which also indicated that itaconate was released from macrophages in a time- or dose-dependent manner. The data from the LC-MS analysis showed that itaconate was released in a similar pattern (Supplementary Fig. 4c, d). The cell supernatant from same pools of cells was tested separately by LC-MS or itaconate biosensor (Supplementary Fig. 4e). The comparable results underscored the reliability and accuracy of our assays (Supplementary Fig. 4e). And according to our assay, when the simultaneous inhibition of MCT1 and MCT4 activity occurred, the amount of secreted itaconate decreased robustly (Fig. 3b and Supplementary Fig. 4f). Su3118, compared to other MCT1/4 inhibitors AZD3965, VB124, AZD0095 or AR-C155858, showed a clear dose-dependent inhibition on the secretion of itaconate (Fig. 3c and Supplementary Fig. 4g–j). We then also evaluated the MCT1 and MCT4-related effect on itaconate concentration by LC-MS. In HEK 293 T cells stably expressing IRG1, overexpression of MCT1 and MCT4 synergistically lowered intracellular itaconate but increased the amount of itaconate in cell supernatant (Fig. 3d and Supplementary Fig. 4k). The inhibition of MCT1 and MCT4 by Su3118 impaired the release of itaconate in RAW264.7 cells (Fig. 3e). Spontaneously, this inhibition led to increasing itaconate in RAW264.7 cells (Fig. 3f). Similarly, in hMDM, itaconate transport is tightly related to MCT1 and MCT4 transport activity (Fig. 3g, h). In *Mct1⁻/⁻Mct4⁻/⁻* RAW264.7 cells, the amount of itaconate also increased intracellularly and decreased in extracellular space (Fig. 3i, j and Supplementary Fig. 4l). Importantly, similar results were obtained when treating *Mct4⁻/⁻* BMDMs with AZD3965 or WT BMDMs with Su3118 (Fig. 3k, l and Supplementary Fig. 4m). The administration of Su3118 in mice also attenuated the secretion of itaconate into the serum or peritoneal lavage supernatant (Supplementary Fig. 4n, o). These data suggested the important roles of MCT1 and MCT4 in transporting itaconate. These data also indicated that the elevated itaconate levels in cells did not lead to increased extracellular itaconate (Fig. 3e, j, k). If the increase in intracellular itaconate results from an upregulated itaconate production rather than a diminished itaconate secretion, more itaconate should have been detected extracellularly. But less itaconate was identified in the supernatant of cells with the compromised MCT1 and MCT4 activity. We further found

that the defect in MCT1 and MCT4 still maintained the level of intracellular itaconate-targeted bacteria even when the production of itaconate was damaged by an IRG1 inhibitor (IRG1-IN-1) (Fig. 3m). These data along with Fig. 2i, j indicated that the high level of intracellular itaconate originates from impaired itaconate secretion caused by deficient MCT1 and MCT4 but not enhanced itaconate production. Given that ABCG2 serves as an exporter of itaconate at plasma membrane, we examined the effect of inhibition of ABCG2, MCT1, and MCT4 on itaconate export. We found that the suppression of MCT1 and MCT4 exerts a more significant influence on itaconate release compared to that of ABCG2 when employing Su3118 or KO143 (an ABCG2 inhibitor) at equivalent concentrations. Importantly, the inhibition of ABCG2, MCT1, and MCT4 synergistically block the release of the majority of itaconate from macrophages (Supplementary Fig. 5a–c). Collectively, these findings have strongly supported that MCT1 and MCT4 are exporters of intracellular itaconate.

### MCT1 and MCT4 transport extracellular itaconate into cells

SLC proteins function as bidirectional transporters depending on the concentration gradient of substrates[23]. We next investigated whether MCT1 and MCT4 also transport extracellular itaconate into cells using itaconate biosensor (Fig. 4a). To avoid the interference of endogenous itaconate, we examined itaconate entry in *Irg1⁻/⁻* cells or HeLa cells, which lack endogenous IRG1 (Supplementary Fig. 1i, 6a). The MCT1 and MCT4 inhibitors Su3118 (Fig. 4b, c) or AZD3965 + VB124 (Supplementary Fig. 6b, c) obviously impaired itaconate uptake into *Irg1*-deficient cell lines. Interestingly, we found that itaconate in the cell supernatant of activated RAW264.7 cells or DC2.4 cells was capable of entering HeLa cells (Supplementary Fig. 6d). And Su3118 or AZD3965 + VB124 hindered itaconate uptake into HeLa cells (Supplementary Fig. 6e). The genetical ablation of MCT1 and MCT4 also impaired itaconate uptake into macrophages (Fig. 4d). Furthermore, less itaconate uptake caused by inhibiting MCT1 and MCT4 transport activity attenuated the effect of itaconate on an oxidative-stress regulator NRF2 (Fig. 4e, f). NRF2 has been reported to be upregulated by itaconate[16] and has a positive role in resisting bacterial infection[47,48]. This implies that itaconate imported by MCT1 and MCT4 also participates in NRF2-dependent antibacterial immunity in cells. More specifically, we utilized ¹³C-labelled itaconate to assess itaconate uptake into BMDMs. We found that Su3118 inhibited the uptake of ¹³C-itaconate in a dose-dependent manner in BMDM cells derived from WT or *Irg1*-deficient mice (Fig. 4g). Considering MCT1 and MCT4 are also responsible for transporting lactate, succinate, and pyruvate, we scrutinized the impact of these substrates on itaconate transport. We observed that itaconate inhibited lactate uptake in a dose-dependent manner. Based on this data, around 20 µM itaconate effectively inhibits 50% of ¹³C-labelled lactate uptake into cells (Fig. 4h). Lactate, succinate, and pyruvate, in contrast to malic acid, also impaired the uptake of itaconate (Fig. 4i). We also constructed plasmids for the inactive mutants of MCT1 and MCT4 as documented for lactate transport[43,49,50]. The results from LC-MS showed that the overexpression of WT MCT1 or MCT4 in cells promoted itaconate transport (Fig. 4j). Furthermore, compared to the MCT1/4 mutants, WT MCT1/4 facilitated more itaconate entry into cells, which subsequently was detected by LC-MS (Fig. 4k, l) or itaconate biosensor (Fig. 4m, n). Consistently, WT MCT1/4 enhanced lactate secretion compared to the MCT1/4 mutants (Supplementary Fig. 6f–k). This suggests that itaconate shares transport pathways with lactate, succinate, and pyruvate via MCT1 and MCT4. Taken together, these findings suggest that MCT1 and MCT4 are transporters responsible for itaconate translocation into cells.

### Itaconate transport through MCT1/4 is pH-dependent

Itaconate is a dicarboxylate with two different ionization status, HITA⁻ and ITA²⁻ ($pK_a$ 3.85 and 5.55)[51]. It will be important to validate whether the variation in protonation states affects itaconate transport. Based on the

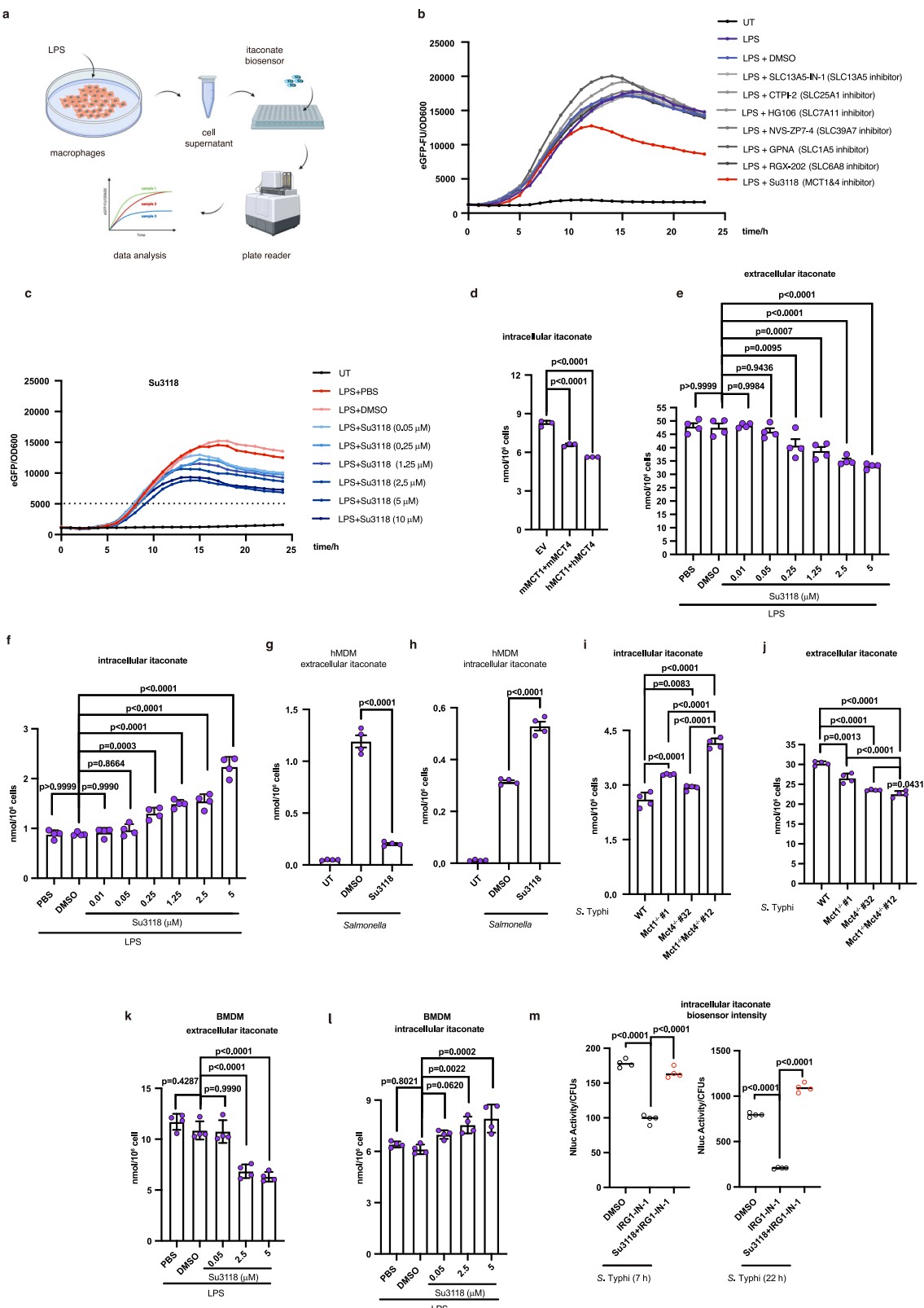

Henderson-Hasselbalch equation, the percentage of monocarboxylic itaconate increases during intracellular acidification (Supplementary Fig. 6l). When stimulating primary macrophages with LPS, we observed a decreased intracellular pH (Supplementary Fig. 6m), which was consistent with the previous report[52]. It indicates that bacterial infection causes intracellular acidification and promotes itaconate to transfer to its monocarboxylate form. To further explore whether the ionization

status of itaconate impacts its transport, we evaluated itaconate uptake and release under different pH conditions. A culture medium containing pH-adjusted itaconic acid was added to the BMDMs. We observed that more itaconate was transported intracellularly at acidic pH (Fig. 4o). We also examined the effect of pH on endogenous itaconate release by adjusting intracellular pH with EIPA. As intracellular pH increased from 5.2 to 8.2, there was a corresponding decrease in the amount of secreted

**Fig. 3 | MCT1 and MCT4 transport itaconate out of cells. a** Development of a biosensor-based assay to detect the secretion of itaconate. **b, c** Detection of itaconate as described in (**a**). RAW264.7 cells stimulated with LPS (100 ng/ml) were treated with the indicated inhibitors for 24 h. The cell supernatant and *S.* Typhi carrying itaconate eGFP biosensor were mixed and cultured in a 96-well plate. The intensity of eGFP was examined. Su3118 (10 μM), RGX-202 (5 μM), GPNA (20 μM), NVS-ZP7-4 (20 nM), SLC13A5-IN-1 (0.044 μM), HG1-6 (5 μM), CTPI-2 (3.5 μM), VB124 (20 μM), AZD3965 (5 μM) in (**b**); Su3118 (indicated) in (**c**). **d** Intracellular itaconate from HEK 293 T cells stably expressing IRG1 transfected with Flag-MCT1 (h, human; m, mouse), Flag-MCT4 (h, human; m, mouse) were measured by LC-MS. **e–l** Itaconate levels were determined by LC-MS in RAW264.7 cells treated with LPS (100 ng/ml) ± Su3118 for 24 h (**e, f**); hMDM treated with Su3118 (5 μM) and infected with *S.* Typhi for 22 h (**g, h**); WT, *Mct1$^{-/-}$*, *Mct4$^{-/-}$*, or *Mct1$^{-/-}$ Mct4$^{-/-}$* RAW264.7 cells infected with S. Typhi for 22 h (**i, j**); and WT or *Mct4$^{-/-}$* BMDM treated with Su3118 and LPS (100 ng/ml) for 24 h (**k, l**). **m** RAW264.7 cells treated with IRG1-IN-1 and Su3118 were infected with *S.* Typhi encoding the nanoluciferase-dependent itaconate biosensor for 22 h. Data are shown as the mean ± SD. Independent biological replicates-*n* = 3 (**d**); *n* = 4 (e-m). A two-tailed Student's *t*-test was conducted for pairwise comparisons (**g, h**). One-way ANOVA was used for multiple comparisons involving a single independent variable (**d–f**, **i–m**). Figure 3a was created in BioRender. Chen, M. (2025) https://BioRender.com/4nf0xdl. Source data are provided as a Source data file.

---

itaconate (Fig. 4p). Consistently, endogenous itaconate accumulated more intracellularly as pH increased (Fig. 4q). Taken together, these data support that itaconate is mainly transported in its monocarboxylate form through MCT1/4.

## Validation of itaconate transport by MCT1/4 in *xenopus* oocytes and liposomes

To gain more convincing data to support that MCT1 or MCT4 directly transport itaconate, we have set up a *Xenopus laevis* oocytes-dependent transport assay (Fig. 5a, b), which is widely employed to investigate monocarboxylate transport[26]. Human cRNA (complementary RNA) of MCT1 or MCT4 was injected into *Xenopus laevis* oocytes and incubated for three days. In the itaconate transport assay, we observed that the oocytes expressing recombinant MCT1 or MCT4 demonstrated enhanced uptake of itaconate (Fig. 5c). This result indicates that MCT1 or MCT4 transports itaconate directly. We next also reconstructed MCT1/4 proteoliposomes by inserting purified human MCT1 or MCT4 protein (Fig. 5d) as described in refs. 22,53. The proteoliposomes were imaged by cryo-TEM or ODT-microscopy (Fig. 5e). Elevated levels of itaconate are efficiently transported into liposomes incorporating WT MCT1 or MCT4, but not those with mutant forms (Fig. 5f, g). These data suggest that MCT1 and MCT4 directly transport itaconate.

## MCT1 and MCT4 transport itaconate into *Salmonella*-containing vacuole

Given that *Salmonella*-containing vacuole (SCV) is developed from the plasma membrane[54], it is possible that MCT1 and MCT4 are also located at SCV and transport itaconate into SCV. To investigate this hypothesis, we first examined whether MCT1 and MCT4 are located at *Salmonella*-containing vacuole. We infected RAW264.7 cells by using *Salmonella* stably expressing mScarlet. MCT1 and MCT4 were detected to be recruited to *Salmonella* (Fig. 6a, b and Supplementary Fig. 7a, b). Moreover, MCT1 and MCT4 bound to intravacuolar *Salmonella* manifested by a SCV-reporter strain (Supplementary Fig. 7c, d)[55], which aligns with our prior observation that itaconate is delivered into SCV[13]. These observations suggested that the association of MCT1/4 and *Salmonella* may have a strong link to itaconate transport at SCV. We then continued to investigate the hypothesis that MCT1 and MCT4 transported itaconate into *Salmonella*-containing vacuole. In this study, our earlier data have shown that itaconate concentration as well as the ratio of itaconate-targeted *Salmonella* were intracellularly improved in macrophages following Su3118 treatment (Figs. 2 and 3). So, we next tried to assess the impact of concentrated itaconate within cells on the proportion of bacteria decorated by MCT1/4. According to our random sampling and statistical analysis, Su3118 increased the percentage of MCT1/4-targeted *Salmonella* (Fig. 6a–d). To validate the necessity of the tight association between MCT1/4 and *Salmonella* in itaconate delivery, we analyzed whether MCT1/4-surrounded *Salmonella* was prone to be targeted by itaconate. We have imaged a random sample of RAW264.7 cells infected by *S.* Typhi carrying itaconate eGFP-biosensor and expressing mScarlet (Fig. 6e, f). Intriguingly, most of bacteria were simultaneously targeted by MCT1/4 and itaconate, which

was enhanced by Su3118 (Fig. 6g, h). And the majority of bacteria decorated by MCT1/4 were targeted by itaconate (Supplementary Fig. 7e-h). Alternatively, for all itaconate-positive bacteria, around 35-40% of them had already been enveloped by MCT1/4 in the control groups treated with DMSO (Fig. 6i, j). Elevating the amount of itaconate by Su3118 triggered a greater number of bacteria (around 60-90%) enveloped by MCT1/4, which were previously untargeted by itaconate (Fig. 6i, j). These data suggest that MCT1 and MCT4 transport itaconate into *Salmonella*-containing vacuole.

## The MCT1 and MCT4-dependent transport of itaconate into SCV is facilitated by RAB32

In consideration of our earlier findings that RAB32, located at mitochondria and SCV[56,57], facilitated the delivery of itaconate into SCV by interacting with IRG1 (Supplementary Fig. 8a)[13,14], the enzyme responsible for itaconate production, we speculated that RAB32 coordinated the MCT1 and MCT4-dependent transport of itaconate. We then examined the association between MCT1/4 and RAB32. According to the co-immunoprecipitation assays, we found that MCT1/4 interacted with RAB32 (Fig. 7a and Supplementary Fig. 8b). Although RAB32 interacts with IRG1 more robustly upon bacterial infection, the association between MCT1/4 and RAB32 has already existed in the absence of bacterial infection (Fig. 7a and Supplementary Fig. 8a), which was not strengthened following *Salmonella* infection (Fig. 7a). More importantly, *Salmonella* typhi was decorated by RAB32 as well as MCT1/4 at the same time (Fig. 7b, c and Supplementary Fig. 8b, c). The co-localization of RAB32 and MCT1/4 probably facilitates the transport of itaconate into SCV. Earlier data have shown that itaconate preferentially targets *Salmonella* strains susceptible to the RAB32-mediated antibacterial effect, such as *Salmonella* Typhimurium ΔGtgEΔSopD2, which lacks GtgE and SopD2. To examine the role of RAB32 in the MCT1 and MCT4-dependent transport of itaconate, we utilized *Salmonella* Typhimurium ΔGtgEΔSopD2 to infect RAW264.7-cells stably expressing eGFP- MCT1/4. The results from microscopy analysis showed that, compared to *Salmonella* Typhimurium, a higher ratio of *Salmonella* Typhimurium ΔGtgEΔSopD2, which cannot neutralize the antibacterial effect of RAB32-itaconate, were targeted by MCT1/4 (Fig. 7d, e, f, g), which underlines that the involvement of RAB32 in mediating MCT1/4-dependent itaconate transport. Collectively, these data suggest, with assistance from RAB32, MCT1 and MCT4 transport itaconate into *Salmonella*-containing vacuole.

## Discussion

We have discovered that the transport function of MCT1 and MCT4 orchestrates the antibacterial immunity of itaconate. As we depicted in Fig. 7h, upon *Salmonella* infection, itaconate generation is catalyzed by IRG1 and then itaconate accumulates intracellularly. MCT4, whose protein level is increased upon infection, along with MCT1 are located at the plasma membrane as well as *Salmonella*-containing vacuole. Comparatively, itaconate is more concentrated in cell cytosol than in the extracellular space or *Salmonella*-containing vacuole early in the infection. So, more itaconate is transported through MCT1 and MCT4 into *Salmonella*-containing vacuole, which is facilitated by RAB32.

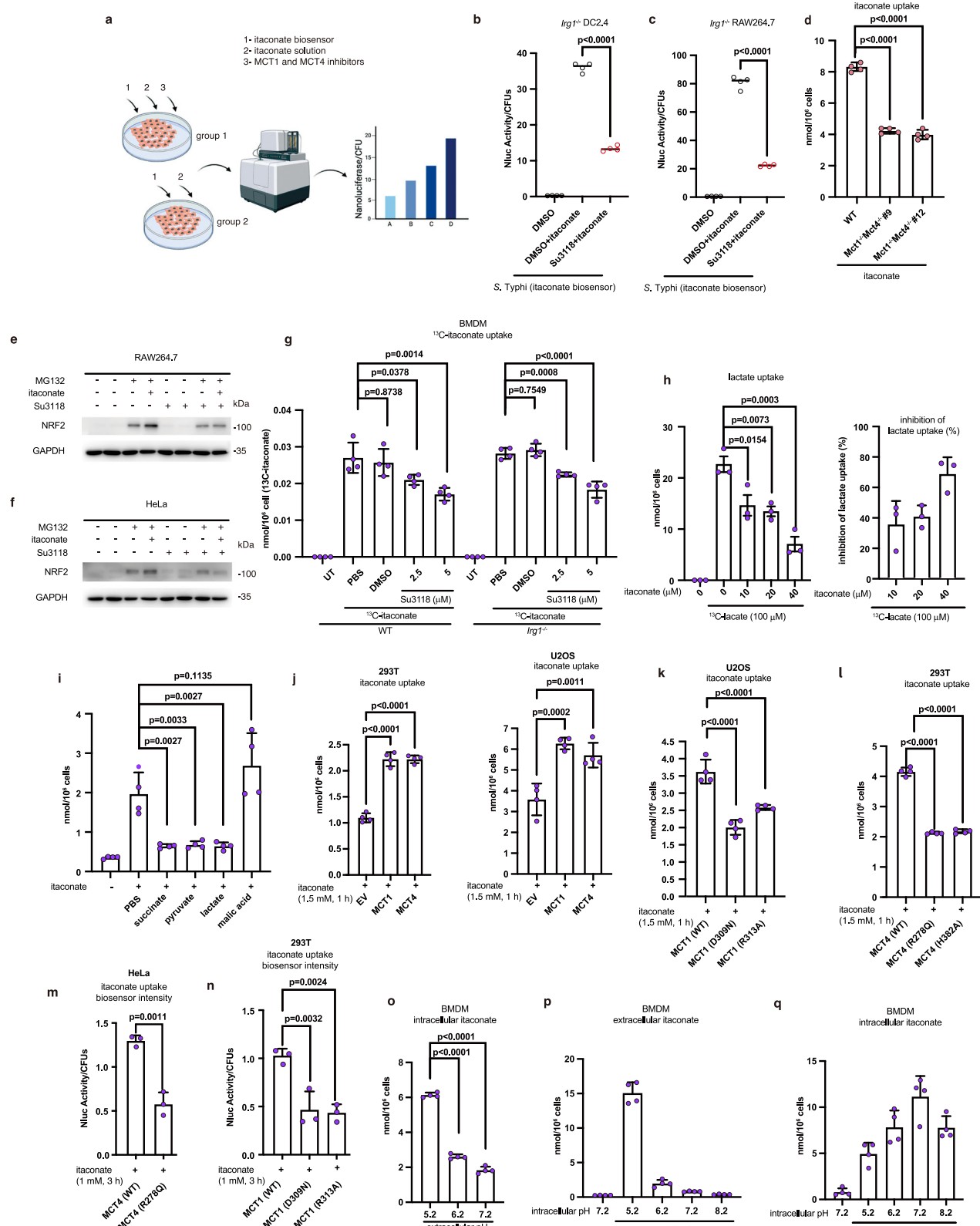

Meanwhile, itaconate is transported out of cells through MCT1 and MCT4 as the infection persists. However, the inhibition of MCT1 and MCT4 by Su3118/AZD3965 + VB124 generally accelerate itaconate accumulation in the intracellular environment, which largely promotes the ratio of itaconate-targeted *Salmonella* and consequently suppresses *Salmonella* survival in macrophages (Fig. 7h). Overall, our findings have confirmed the significance of modulating MCT1 and

MCT4-dependent itaconate transport during bacterial infection. And the antibacterial immunity of itaconate is exaggerated in macrophages defective in MCT1 and MCT4 transport activity. But itaconate accumulation resulting from Su3118 treatment may not be sufficient on its own and may require co-stimulatory signals to modulate lysosomal biogenesis or the polarization of macrophages. And, besides itaconate, other substrates of MCT1 and MCT4, which accumulates intracellularly

**Fig. 4 | MCT1 and MCT4 transport itaconate into cells. a** Development of a biosensor-based assay to detect itaconate uptake. **b–d** *Irg1*[-/-] DC2.4 or RAW264.7 cells treated with itaconate (4 mM) ± inhibitors were infected with *S*. Typhi carrying the itaconate nanoluciferase biosensor for 22 h, and nanoluciferase activity was measured; itaconate uptake in WT and *Mct1*[-/-], *Mct4*[-/-], or *Mct1*[-/-]*Mct4*[-/-] RAW264.7 cells (3 h) was analyzed by LC-MS. RAW264.7 (**e**) or HeLa (**f**) cells were treated with MG132 (10 μM), itaconate (1 mM), or Su3118 (10 μM) as indicated. The cell lysates were analyzed by the immunoblotting with anti-NRF2 and anti-GAPDH. **g** WT, *Irg1*[-/-] BMDM were incubated with Su3118 (4 h, 2.5 μM, 5 μM) and [13]C-itaconate (2 h, 50 μM). The level of [13]C-itaconate was determined by LC-MS. **h** *Irg1*[-/-] RAW264.7 cells were incubated with itaconate (0–40 μM) and [13]C-lactate (100 μM) for 1 h; lactate uptake was measured by LC-MS (Left), and inhibition by itaconate was calculated (Right). **i** BMDM were incubated with itaconate, succinate, lactate, or malic acid (0.5 mM, 0.5 h), and itaconate uptake was measured by LC-MS. HEK293T (**j**, **m**, **l**), U2OS (**j**, **k**), or HeLa cells (**n**) transfected with EV, MCT1, MCT4, or their mutants were incubated with the indicated itaconate, and uptake was measured by LC-MS or using the itaconate biosensor in *S*. Typhi. **o** BMDM were incubated in medium adjusted to pH 5.2–7.2 with itaconate (1 mM, 2 h), and uptake was measured by LC-MS. **p**, **q** BMDM were adjusted to pH 5.2–8.2 using EIPA and treated with LPS (100 ng/ml, 8 h). Itaconate levels were determined by LC-MS. Data are shown as the mean ± SD. Independent biological replicates-*n* = 3 (**h**, **m**, **n**); *n* = 4 (**b–d**, **g**, **i–l**, **o–q**). A two-tailed Student's *t*-test was conducted for pairwise comparisons (**b**, **c**), while one-way ANOVA was used for multiple comparisons involving a single independent variable (**d**, **g–q**). Figure 4a was created in BioRender. Chen, M. (2025) https://BioRender.com/4nf0xdl. Source data are provided as a Source data file.

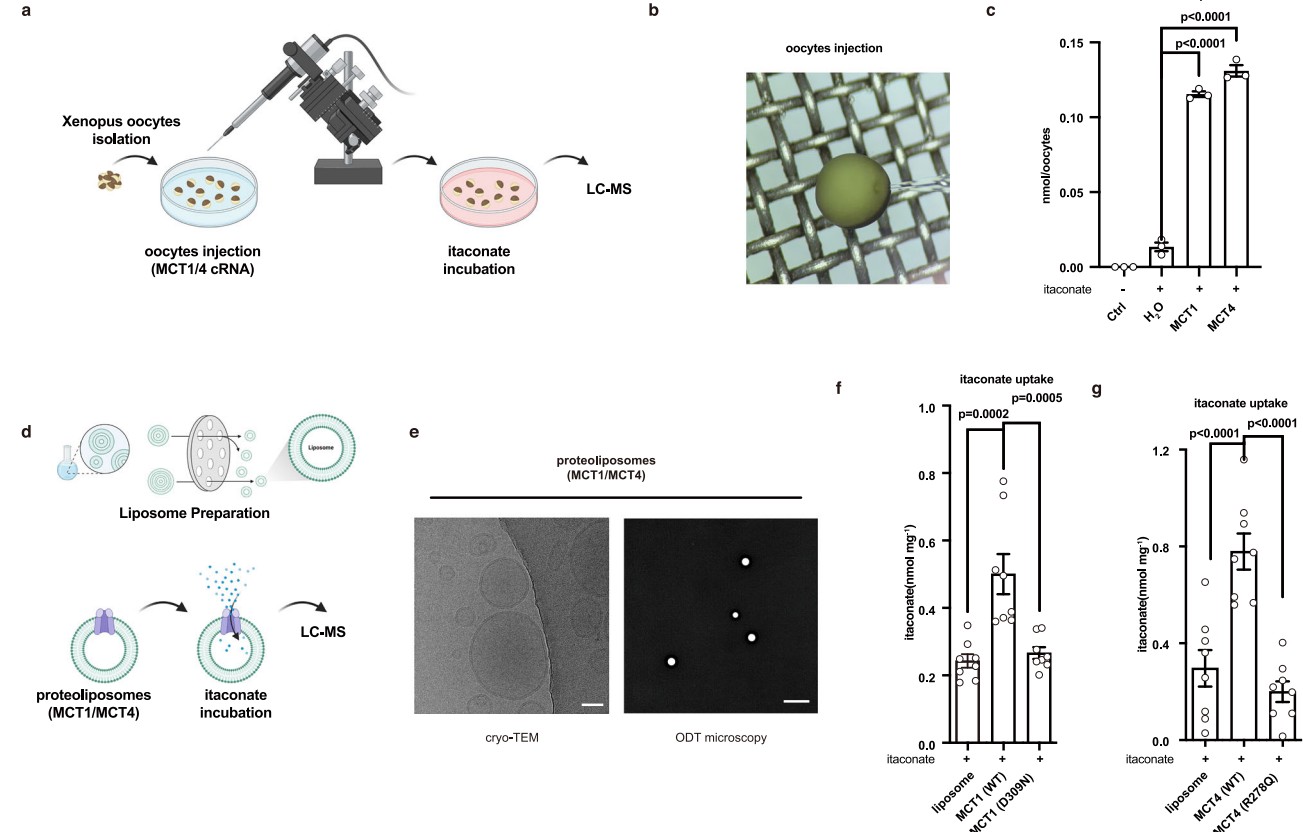

**Fig. 5 | Validation of itaconate transport by MCT1/4 in xenopus oocytes and liposomes. a** Schematic illustrating itaconate transport into *Xenopus laevis* oocytes mediated by MCT1/4. **b** A representative image of a *Xenopus laevis* oocyte during injection. **c** Itaconate (1 mM) was incubated with *Xenopus laevis* oocytes expressing human MCT1 or MCT4 for 10 min. The uptake of itaconate was measured by LC-MS. **d** Schematic illustrating itaconate transport mediated by MCT1/4 into liposomes. **e** Images of proteoliposomes were captured from Cryo-TEM (left) and MH-HoliView panoramic super-resolution microscope (right). Scale bars indicated 50 nm and 10 μm, respectively. **f**, **g** The LC-MS analysis of itaconate uptake following the incubation of liposomes reconstituted with MCT1/4 protein and itaconate (1 mM) for 1 h. The elution buffer containing an equal amount of HA peptides was used to generate control proteoliposomes. Data are shown as the mean ± SD. Independent biological replicates-*n* = 3 (**c**); *n* = 8 (**f**, **g**). One-way ANOVA was used for multiple comparisons involving a single independent variable (**c**, **f**, **g**). Figure 5a, d was created in BioRender. Chen, M. (2025) https://BioRender.com/4nf0xdl. Source data are provided as a Source data file.

under Su3118 treatment, probably also contribute to this outcome, which requires further study.

We have also confirmed that MCT1 and MCT4 transported extracellular itaconate into cells (Fig. 4). The attenuation of the itaconate-related effect was noted when inhibiting its uptake via MCT1 and MCT4, as evidenced by the downregulation of NRF2 expression (Fig. 4e, f). Itaconate, imported through MCT1 and MCT4, may act as an important messenger to activate antibacterial activity or other immune responses intracellularly. Additional research is necessary to comprehensively understand the physiological effect of itaconate transported by MCT1 and MCT4. Besides, although the concentration of itaconate in cell supernatant in vitro is not high enough to exert antibacterial activity, it would be not surprising that the secreted itaconate fulfills the antibacterial immune responses in vivo. For instance, the intestinal epithelial cells facing bacterial infection may uptake itaconate through MCT1/4 in the intestinal lumen to resist bacteria.

Comparing the itaconate transport activity of MCT1 and MCT4 (Figs. 3 and 4), we found that MCT1 and MCT4 are essential for itaconate transport. In contrast to the effects of genetic depletion of MCT1 or MCT4 alone on itaconate transport, the simultaneous depletion of both MCT1 and MCT4 resulted in a more pronounced reduction in itaconate transport. Furthermore, overexpression of

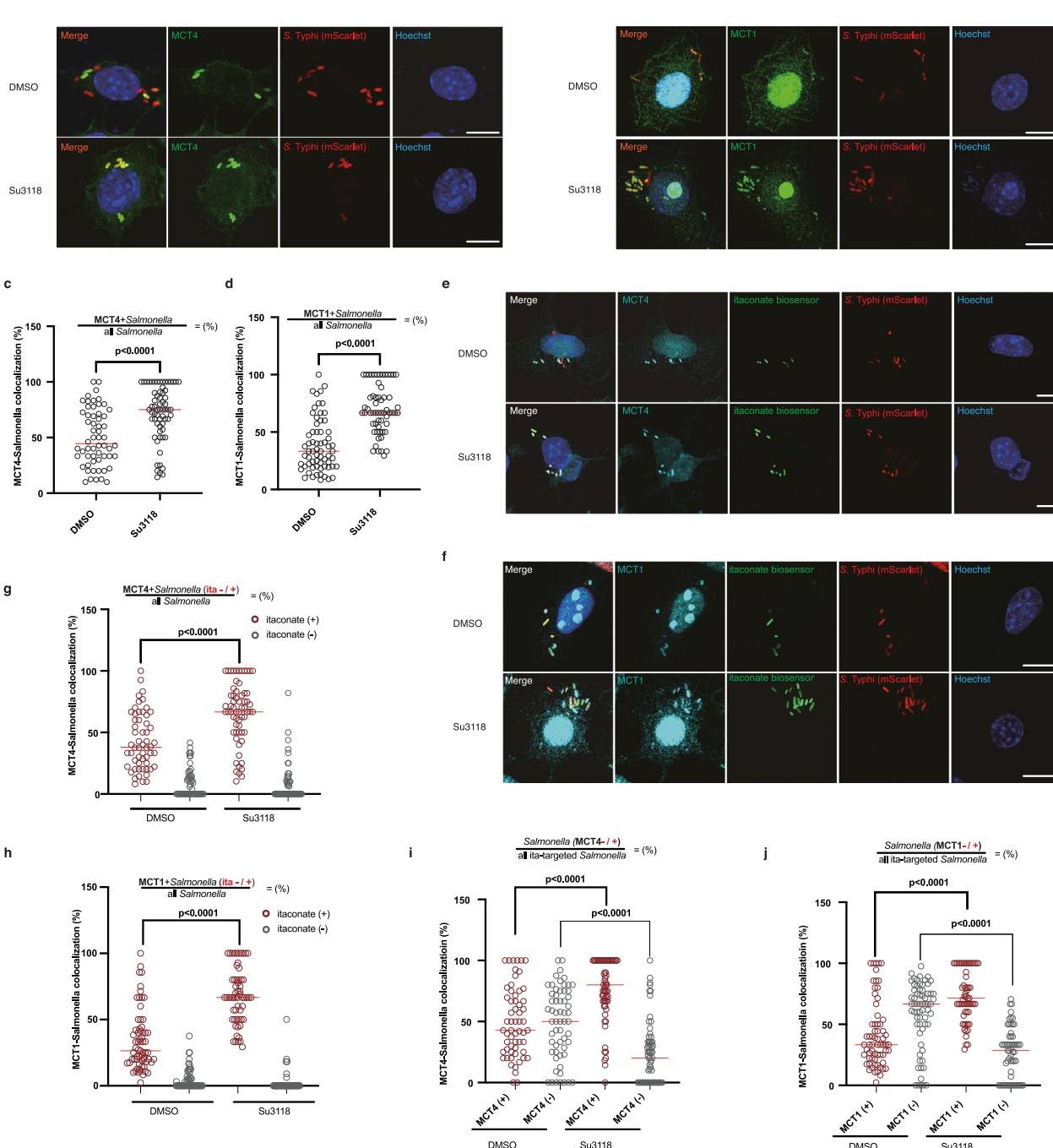

**Fig. 6 | MCT1 and MCT4 transport itaconate into *Salmonella*-containing vacuole. a–d** RAW264.7 cells treated with DMSO/Su3118 and infected by *S.* Typhi carrying a plasmid encoding mScarlet (Red) for 16 h were fixed, stained with Hoechst (Blue) to mark nuclei, and stained with an anti-MCT1/4 along with Alexa 488-conjugated anti-rabbit antibody (Green), and imaged under a fluorescence microscope. Around 60-70 cells were collected randomly. Representative views of RAW264.7 cells were displayed (**a**, **b**). The ratio of bacteria decorated by MCT1/4 in was quantified: % = $\frac{\text{MCT}+Salmonella}{\text{All }Salmonella}$ (**c**, **d**). Each dot represents the ratio of MCT1/4-decorated bacteria within one cell. **e–j** RAW264.7 cells treated with DMSO/Su3118 and infected by *S.* Typhi carrying plasmids encoding mScarlet (Red) and itaconate eGFP biosensor (Green) for 16 h were fixed, incubated with Hoechst (Blue) and MCT1/4 antibodies along with Alexa 647-conjugated anti-rabbit antibody (Cyan), and imaged. More than 60 cells were collected unbiasedly. Representative views of

RAW264.7 cells were displayed (**e**, **f**). **g**, **h** In each cell, we counted the total number of *Salmonella*, the number of MCT1/4-undecoraed/decorated *Salmonella* (MCT1/4 −/+) and the number of itaconate-untargeted/targeted-*Salmonella* (ita −/+). Then the ratio of itaconate-untargeted/targeted (ita −/+) *Salmonella* (already decorated by MCT1/4) was calculated: % = $\frac{\text{MCT}+Salmonella(\text{ita}-/+)}{\text{All }Salmonella}$. **i**, **j** The proportion of MCT1/4-undecoraed/decorated *Salmonella* (MCT1/4 −/+) in a pool of itaconate-targeted *Salmonella* was calculated: % = $\frac{Salmonella(\text{MCT}-/+)}{\text{All ita}-\text{targeted }Salmonella}$. Each dot represents the indicated ratio within one cell. Scale bar, 10 μm. Independent biological replicates-*n* = 58 (**c**, DMSO; **g**, DMSO), 60 (**c**, Su3118), *n* = 65 (**d**, DMSO; **h**, DMSO; **j**, DMSO), *n* = 59 (**d**, Su3118; **h**, Su3118; **i**, DMSO; **j**, Su3118), *n* = 64 (**g**, Su3118; **i**, Su3118). A two-tailed Student's *t*-test was conducted for pairwise comparisons (**c**, **d**, **g**, **h**, **i**, **j**). Source data are provided as a Source data file.

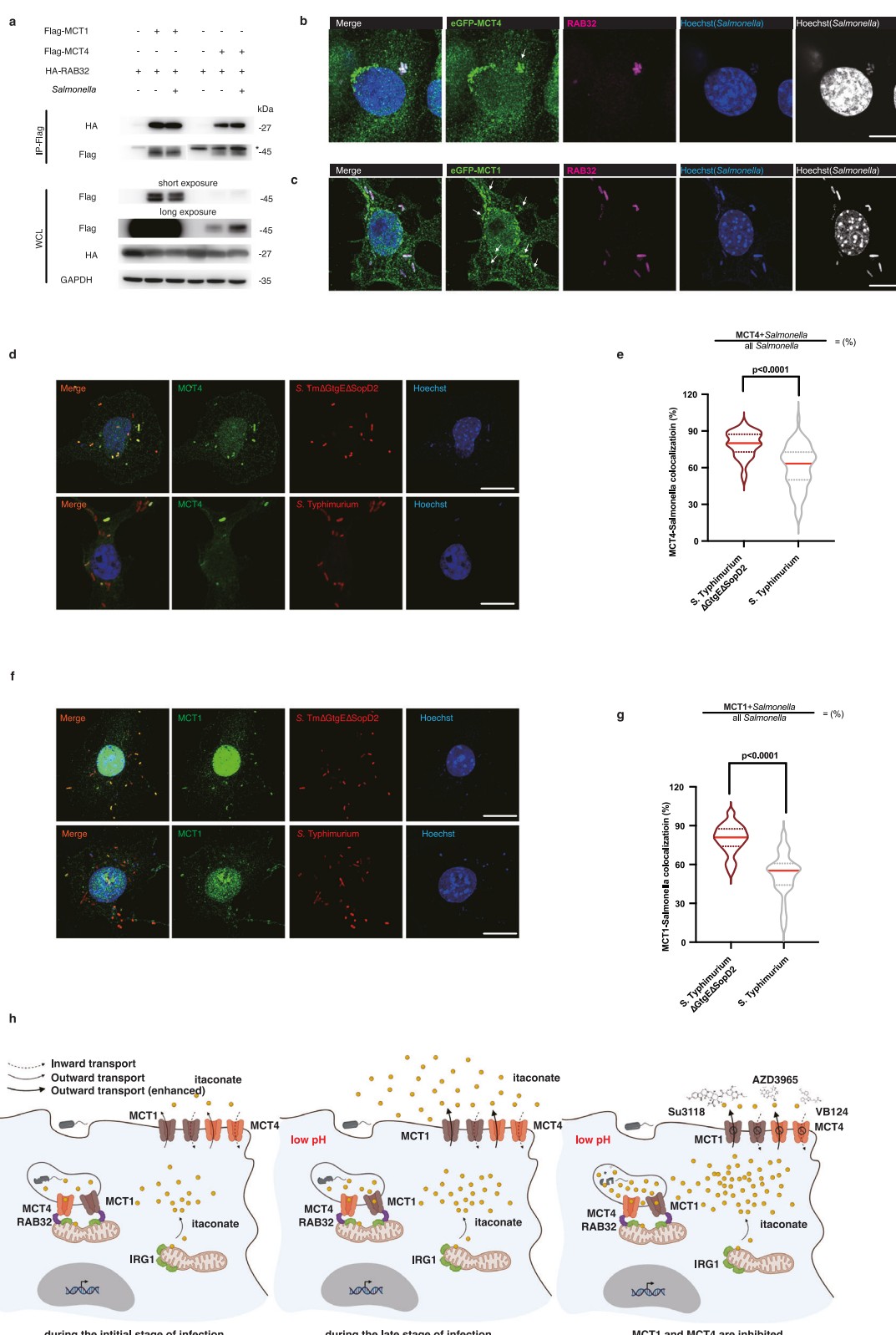

either MCT1 or MCT4 enhanced itaconate transport. These data suggest that MCT1 and MCT4 may functionally compensate for each other in itaconate transport under certain conditions. Additionally, MCT1 is a widely expressed protein, found in the macrophages as well as various mouse tissues such as spleen, liver, brain, and kidney, while MCT4 is an inducible protein in the macrophages, spleen and liver, but not expressed in the brain and kidney (Fig. 1o, p, and Supplementary

Fig. 1o). These observations suggest that MCT1 and MCT4 may function differently in the uptake of itaconate across different tissues. Given that both transporters are also known to carry lactate or other related substrates[41,43], their efficiency and specificity for itaconate transport may be influenced by substrate competition, depending on the cellular context and metabolic demands. Their transport activity is probably not fixed but dynamically regulated by the availability of

**Fig. 7 | The MCT1 and MCT4-dependent transport of itaconate into SCV is facilitated by RAB32. a** HEK293T cells transfected with HA-RAB32 and Flag-MCT1/4 were analyzed by immunoblotting and co-immunoprecipitation with anti-HA, Flag, and GAPDH antibodies. **b, c** eGFP- MCT1/4 stable RAW264.7 cell lines infected by *S*. Typhi for 22 h were fixed, stained with Hoechst (Blue) and RAB32 (Magenta) antibody along with Alexa 647-conjugated anti-rabbit antibody, and imaged. Scale bar, 10 μm. **d–g** RAW264.7 cells were infected by *S*. Typhimurium or *S*. TyphimuriumΔGtgEΔSopD2 carrying a plasmid encoding mScarlet (Red) for 22 h and fixed, stained with Hoechst (Blue) to mark nuclei, and incubated with an anti-MCT1/4 along with Alexa 488-conjugated anti-rabbit antibody (Green), and imaged under a fluorescence microscope. More than 60 cells were collected unbiasedly. Representative views of RAW264.7 cells were displayed in 7 **d** and 7 **f**. In each cell, we counted the total number of *Salmonella*, the number of MCT1/4-undecoraed/decorated *Salmonella* (MCT -/+). Then the ratio of MCT1/4-decorated *Salmonella* was calculated: $\% = \frac{Salmonella + MCT1/4}{All Salmonella}$ (**e, g**). **h** A working model for the mechanism by which MCT1 and MCT4 transport itaconate (Created in BioRender. Chen, M. (2025) https://BioRender.com/4nf0xdl). Scale bar, 10 μm. Independent biological replicates-n = 60 (**e, g**). A two-tailed Student's t-test was conducted for pairwise comparisons (**e, g**). Source data are provided as a Source Data file.

competing molecules. We have discovered that the presence of itaconate attenuated lactate uptake (Fig. 4h). And lactate also reduced itaconate uptake (Fig. 4i). Since MCT1 has a higher affinity for lactate compared to MCT4[41], it is possible that lactate competition has a greater impact on itaconate transport through MCT1. Further investigations along these lines would be valuable in the future.

Our finding indicates that the environmental pH affects the transport of itaconate through MCT1 and MCT4. Due to protonation, itaconate exists in different forms: monocarboxylate (HITA⁻) and dicarboxylate (ITA²⁻)[51]. The ratio of monocarboxylate (HITA⁻) form is increasing due to the acidification of cells in response to infection[52] and LPS treatment (Supplementary Fig. 6m). This property may contribute to the accumulation of itaconate during the early stage of infection to restrict intracellular bacteria and facilitate the export of excess itaconate for a paracrine signal as cells experience acidification during the late stage of infection. A similar mechanism has been investigated in detail for the dicarboxylate succinate, where pH-gated release of succinate occurs in response to exercise[26].

The locations of MCT1/4 at the cell membrane and cytosolic SCV were detected at the same time (Figs. 2 and 6). But little known is about whether the MCT1/4 inhibitors used here are absorbed into cells and directly inactivate intracellular MCT1/4. So far, no experimental data have shown that VB124 and Su3118 are membrane permeable, though AZD3965 has been reported to enter cells[58]. According to our data, these inhibitors increased intracellular itaconate concentration in general, which also generally induced itaconate-derived bactericidal activity. Thus, at least, our study has suggested that the inhibitors of MCT1/4 mainly impair the transport of itaconate across plasma membrane but not SCV.

Importantly, our findings have provided evidence that itaconate transport is a key process in cell-autonomous defense against invading bacteria. Considering itaconate as a vertical metabolite, the discovery of itaconate transporters will also provide important clues for further study in different research fields. Besides, our findings connect the antibacterial responses with MCT1 and MCT4, the well-known druggable targets, which may advance the development of antimicrobial drugs along with the therapeutic potential of itaconate itself. The antibacterial properties of MCT1 and MCT4 inhibitors (i.e., Su3118, VB124 + AZD3965) endow them with newfound significance, particularly crucial in the fact of the threat of antibiotic failures in treating bacterial infections.

## Methods

### Mice

The animal care and experimental protocols underwent thorough review and received the approval from the Institutional Animal Care and Use Committee at Shenzhen Bay Laboratory (SZBL). Mice were maintained in a pathogen-free facility with a 12-h light/12-h dark cycle, and had free access to food and water. The housing temperature was controlled at approximately 22 °C, with relative humidity around 50–60%. Mice were sacrificed via CO₂ inhalation, following the protocols established by the Institutional Animal Care and Use Committee at Shenzhen Bay Laboratory. Mice experiments included both sexes, with age- and sex-matched pairs for each genotype in all conditions.

*Irg1⁻/⁻* mice (C57BL/6NJ-Acod1<em1(IMPC)J > /J; stock No: 029340) were purchased from The Jackson Laboratory. *Mct4⁺/⁺* mice (#S-KO-15319) were purchased from Cyagen. *Mct4* homozygous mice were obtained by breeding *Mct4* heterozygous mice (#S-KO-15319) and confirmed through genotyping. LB broth containing *Salmonella* strains was cultured overnight at 37 °C and diluted 1:20 in LB supplemented with 0.3 M sodium chloride (NaCl), which was then cultured until reaching an OD600 of 0.9. Intraperitoneal (IP) infection of 8-to-12-week-old animals was carried out using different *Salmonella* Typhimurium strains (CFU as indicated in the figure legends). Mice were administered intraperitoneal injections of Su3118 (5 mg/kg) or AZD3965 (100 mg/kg), as previously documented[36,41,59]. Corn oil (Aladdin #C116025) was utilized as a solvent for both Su3118 and AZD3965, in accordance with the manufacturer's chemical guidelines. Mice receiving intraperitoneal injections of corn oil served as control groups for the Su3118 and AZD3965 treatments. Itaconate in serum or peritoneal lavage supernatant of mice infected was collected and measured as described below. To assess bacterial burdens in the spleens, mice were euthanized at designated time points post-infection, as indicated in the figure legends, and their organs were homogenized in 2 ml of PBS containing 0.1% sodium deoxycholate (0.1% DOC). Subsequent dilutions were plated onto LB agar plates for quantification of colony-forming units (CFUs). To assess the expression of IRG1/MCT1/4/GAPDH, multiple tissue samples from mice were isolated, homogenized, and subjected to subsequent procedures for western blot analysis described below.

### Cell culture and Transfection

RAW264.7 (#SCSP-5036) and HeLa cells (#TCHu187) were obtained from National Collection of Authenticated Cell Cultures. HEK 293T cell line was kindly provided by Professor Gong Cheng (Tsinghua University). DC2.4 cells, as kindly provided by Professor Xun Sun (West China School of Pharmacy, Sichuan University). Human monocyte-derived macrophages (hMDM) were kindly provided by Professor Qiankun Wang (Shenzhen Bay Laboratory). Briefly, anonymous human cord blood samples were acquired from MILECELL BIO. To obtain monocyte-derived macrophages, CD14⁺ cells were cultured for 6–7 days in the presence of 50 ng/ml recombinant human GM-CSF and 20 ng/ml recombinant human M-CSF. The RAW264.7 and HeLa cell lines were cultured in Dulbecco's modified Eagle medium (DMEM, GIBCO) supplemented with 10% FBS (#FBS-E500, Newseum) at 37 °C with 5% CO₂ in a humidified incubator. All cell lines were monthly tested for mycoplasma contamination. BMDC (bone marrow-derived dendritic cells) or BMDM (Bone marrow-derived macrophages) were isolated from 8-to-12-week-old mice. BMDC or BMDM were cultured with granulocyte-macrophage colony-stimulating factor (GM-CSF)-containing media or the collected supernatant of L929 cells for around 6–7 days before infection, respectively.

Mammalian cells were treated with the indicated reagents using the specified working concentrations as below, unless otherwise specified in the figure legends: itaconic acid (4 mM), Su3118 (10 μM), AZD3965 (1 or 5 μM), VB124 (20 μM), RGX-202 (5 μM), GPNA (20 μM), NVS-ZP7-4 (20 nM), SLC13A5-IN-1 (0.044 μM), HG1-6 (5 μM), CTPI-2 (3.5 μM), IRG1-IN-1 (1 mM), MG132 (10 μM), puromycin (2 μg/ml to 10 μg/ml), blasticidin (2 μg/ml to 10 μg/ml). Unless is indicated in the

figure legends, pH of all reagent solutions is not adjusted. In a transfection assay, HEK 293T cells were seeded into 6/12/24-well plates and incubated for around 18 h. HEK 293T cells were transfected by following the Lipofectamine 2000 instructions and collected for the indicated analysis in figure legends.

## Construction of CRISPR/Cas9 knockout cell lines
The knockout Cell lines were generated by following the CRISPR/Ca9 genetic editing protocol[60]. Simply, pseudotyped virus was generated by co-transfecting pVSVG, Δ8.9, and lentiCRISPR v2 (gRNA) plasmids at a ratio of 3:2:5 into HEK-293 cells in a 6-well plate by Lipofectamine 2000. Cell culture supernatants were harvested 24 h post-transfection and utilized to transduce the indicated cell lines. Puromycin or blasticidin was used to select and maintain the knockout cell lines. For genotyping of the knockout cell lines, we extracted genomic DNA from cells using a DNA isolation kit (DP304-03, TIANGEN) as DNA templates and subjected to PCR. The PCR products were sequenced and analyzed. Sequences for the gRNAs used are 5'-TATAGCGTGTCGAAGCTTGG-3' for $Irg1$, 5'- GTAG TAGCGCCCATAGATAG -3' for $Mct1$, 5'- TGGCCGTAGTTCCGGGGGCT -3' for $Mct4$. To construct IRG1 or MCT1/4 stable cell lines, similar procedures to those described above were followed. However, pseudotyped virus was generated by co-transfecting pVSVG, Δ8.9, and lenti_CMV_gene into HEK-293 cells in a 6-well plate at a ratio of 3:2:5 using Lipofectamine 2000. Selection and maintenance of the stable cell lines were also achieved using puromycin or blasticidin.

## Itaconate transport assay in Xenopus laevis oocytes
Xenopus laevis oocytes were collected following a well-established protocol[61]. The transport assay was conducted as described in refs. 26,61. Oocytes were injected with MCT1 or MCT4 cRNA (25 ng/oocyte) transcribed from the human MCT1 or MCT4 cloned in the T7-containing plasmid pcDNA3.1 using the mMessage mMachine kit (Thermo Fisher Scientific #AM1344). Oocytes injected with equivalent volume of water acted as our experimental controls. Oocytes were cultured for 3 days in modified Barth's solution (78 mM NaCl, 1 mM KCl, 0.33 mM Ca(NO$_3$)$_2$, 0.41 mM CaCl$_2$, 0.82 mM MgSO$_4$, 2.4 mM NaHCO$_3$, 10 mM HEPES sodium salt, 1.8 mM sodium pyruvate, 10 μg/mL gentamycin, 10 μg/mL streptomycin, pH 7.6). To evaluate itaconate uptake, groups of 15 oocytes were incubated in OR2 buffer (2.5 mM NaCl, 2.5 mM KCl, 1 mM CaCl$_2$, 1 mM MgCl$_2$, 1 mM Na$_2$HPO$_4$, 5 mM HEPES, pH 6.4) containing itaconate for 10 min. Oocytes were lysed with methanol-acetonitrile-water after they are washed in ice cold OR2 buffer, and subjected to LC-MS analysis, as described below.

## Liposomes preparation
Liposomes were reconstituted following a previously described protocol with minor adjustments[22,53]. DOPC, DOPE, and DOPG were mixed in a 1:1:1 ratio (20 mg total), and 2 mg of cholesterol was added before dissolving the mixture in 5 mL chloroform. The lipids were dried in a rotary evaporator at 40 °C with gentle rotation (40 rpm) for 2 h, then re-dissolved in 20 ml dichloromethane and incubated at 30 °C for 30 min in a rotary evaporator. The lipid mixture was dissolved in 50 mM KPi buffer (KH$_2$PO$_4$ + K$_2$HPO$_4$, pH 7.0) and sonicated at 70 W (2 s on, 4 s off) for 10 min. After three freeze-thaw cycles using liquid nitrogen and liposomes were generated by extrusion through a 400 nm polycarbonate filter (Whatman Inc.) 11 times using a mini-extruder (LiposoFast, Avestin Inc.). To ensure complete liposome solubilization, the suspension was titrated with continuously 10% Triton X-100 at a ratio of 1:500. When the optical density at 540 nm (OD540) reached its maximum, five additional aliquots of Triton X-100 were added for complete solubilization of liposomes.

## Proteoliposomes reconstitution
40 μg pcDNA3.1-HA-tagged MCT1/4 plasmids were transfected into EXPI293 (Thermo) or HEK 293 T cells in T175 flasks. After 48 h, cells were lysed in Triton X-100 buffer (50 mM Tris-HCl, pH 7.4, 150 mM NaCl, 5% glycerol, and 1% Triton X-100), and proteins were enriched using 200 μL Anti-HA Affinity Gel (Sigma). Following four washes, proteins were eluted by 200 μL of 100 μg/ml HA peptides (Sigma) in buffer A at 4 °C for 1 h (50 mM KPi, 20% glycerol, 200 mM NaCl, 500 mM imidazole, 0.05% DDM, pH 7.8). The purified proteins and liposomes were combined at a 1:100 (wt/wt) ratio. Detergents were removed by incubating the sample with 500 mg/mL Bio-Beads SM-2 (Bio-Rad) overnight at 4 °C, ensuring that air space did not exceed 10% of the tube volume, and the resulting proteoliposomes were collected from the flow-through of polyprep columns (Bio-Rad). Finally, the proteoliposomes were concentrated to 5-10 mg/ml in buffer T (25 mM HEPES, pH 7.5, 150 mM NaCl, and 5 mM MgCl$_2$) by ultracentrifugation at 270,000 × $g$ for 2 h at 4 °C with optima XPN-80 (Beckman).

## In vitro Itaconate transport assay
For the in vitro Itaconate transport assay, proteoliposomes were incubated with 1 mM itaconic acid in a total of 100 μL buffer T for 1 h at room temperature. Afterward, the liposomes were washed twice with 1 ml of ice-cold buffer T, and metabolites inside the liposomes were extracted using 500 μL of methanol:ACN:H$_2$O (2:2:1). The solvent was then evaporated to dryness at 4 °C overnight. The dried metabolites were dissolved in 30 μL of 50% ACN, and 1 μL of the solution was analyzed using a QTRAP 6500 LC-MS/MS system.

## Cyro TEM
Sample preparation for Cryo-TEM was performed using the Vitrobot IV, with humidity maintained at 100% and the temperature set to 10 °C throughout the experiments. 300-mesh copper grids coated with carbon film (Quantifoil® R 1.2/1.3 300 Mesh, Cu) were glow-discharged using a Pelco EasiGlow unit to render the carbon film hydrophilic. Then, 3 μL aliquots of the proteoliposomes were applied to each grid, followed by a 10-second adsorption time. The grid was manually blotted for 3 s using Ted Pella filter paper before being plunged into liquid ethane, which was cooled by liquid nitrogen. The frozen grids were stored in liquid nitrogen until use. Samples were examined with a Tundra microscope operating at 100 kV, with an electron dose of 40 electrons/Å for imaging. Images were captured using a CETA F camera.

## Panoramic super-resolution microscopy
Proteoliposomes were seeded onto the glass bottom of 3.5 cm dishes (NEST), and images were captured using optical diffraction tomography (ODT) with a live MH-HoliView Panoramic super-resolution microscope (Pellucid Optics Technology, Nantong Co., LTD). The images were then processed using Fiji software to enhance visualization.

## Bacterial strains
Salmonella enterica subsp. enterica serovar Typhimurium str. SL1344 was kindly provided by Professor Feng Shao (National Institute of Biological Sciences). Salmonella enterica subsp. Enterica serovar Typhi was purchased from National Center for Medical Culture Collections (CMCC#50071).

## Salmonella genetic deletion strains
Salmonella Typhimurium ΔGtgE ΔSopD2 was constructed by using standard recombinant DNA and allelic exchange procedures in which E. coli SM10 (λπ) acts as the conjugative donor strain. Select merodiploids by plating diluted bacteria on LB plate containing chloramphenicol (50 μg/ml). The merodiploids were selected and examined by PCR. One merodiploid was picked and incubated in LB broth at 37 °C for 1 h. The recombinants were selected on LB agar plates (no NaCl) supplemented with 10% sucrose at 30 °C overnight. The mutant strain was carefully confirmed by PCR and DNA sequencing and investigated by the infection procedures.

## Plasmids, antibodies and reagents

All plasmids were constructed by polymerase chain reaction, restriction enzymes digestion, DNA ligation, or Gibson assembly. The plasmids used for labelling bacteria were constructed by placing mScarlet into pWSK129 vector. The plasmid for *Salmonella*-containing vacuole (SCV) reporter strain was constructed by cloning mScarlet and the pltB promoter sequences into pBAD24 vector referred to a previous study[55]. The plasmid for constructing itaconate biosensor (eGFP or nanoluciferase) was constructed and prepared according to the previous study[13]. For detailed plasmid information, please refer to Table S1.

Antibodies used were anti-MCT1 (GXP457728, genxspan; Novus, NBP1-59656), anti-MCT4 (22787-1-AP, proteintech), anti-IRG1 (19857S, CST), anti-NRF2 (12721S, CST), anti-β-actin (A1978, sigma), anti-GAPDH (5174S, CST), anti-RAB32 (GTX130477, Genetex), anti-lamp1 (ab208943, abcam), anti-Flag M2 (F1804, Sigma), anti-HA (26183, Thermo), Goat anti-Mouse IgG (H + L) Secondary Antibody (C31430100, thermos), Goat anti-Rabbit IgG (H + L) Secondary Antibody (C31460100, thermos). 488-conjugated Goat Anti-Mouse IgG(H + L) (SA00013-1), 488-conjugated Goat Anti-Rabbit IgG(H + L) (SA00013-2), 594-conjugated Goat Anti-Mouse IgG(H + L) (SA00013-3); 594-conjugated Goat Anti-Rabbit IgG(H + L) (SA00013-4) antibodies were from proteintech.

Other reagents used in our study are listed below. Itaconic acid was purchased from Sigma (#I29204-100G). LCMS-grade Acetonitrile (#1.00029.2508) and methanol (#1.06035.2500) were purchased from Merck. SLC inhibitors Su3118 (S9907), AZD3965 (S7339), VB124 (S9929), RGX-202 (E1223), GPNA (S6670), NVS-ZP7-4 (S6668), SLC13A5-IN-1 (S0755), HG1-6 (E1080), CTPI-2 (S2968), 5-(N-Ethyl-N-isopropyl)-Amiloride (EIPA) (S9849) were purchased from Selleck. IRG1-IN-1 (HY-148335) was purchased from MCE. DAPI (C1002, Beyotime) or Hoechst stain (H1398, Thermo) used for DNA/cell nucleus staining. $^{13}$C-labeled itaconate (0.5 mg, sc-495554) was purchased from SANTA. Lipofectamine 2000 (11668019, Thermo) was used for transfection. LPS of bacteria is purchased from Sigma (L2018). GM-CSF (315-03) for culturing BMDC is purchased from PEPROTECH. Puromycin (ant-pr-1) and blasticidin (ant-bl-05) were purchased from InvivoGen.

## Biosensor-based itaconate detection assay

*Salmonella* strains carrying plasmids encoding the nanoluciferase or eGFP itaconate biosensors were cultured overnight for around 18 h. To detect itaconate secretion, the overnight culture of *Salmonella* carrying the eGFP itaconate biosensor was diluted to an OD600 of 0.05 to 0.1 in LB and grown until the OD600 reaching 0.3. The *Salmonella* cultures were centrifuged and resuspended into PBS. The mammalian cell supernatant was added to the *Salmonella* cultures, and seeded into a 96-well plate. The expression of the eGFP was measured by the Tecan Infinite®M1000 plate reader. To detect the level of intracellular itaconate, the overnight culture of *Salmonella* carrying the eGFP/nanoluciferase itaconate biosensor were sub-cultured and infected mammalian cells by following the described bacterial infection protocol. eGFP was detected by microscope as described in this study. Luciferase intensity of the itaconate biosensor was measured by using the Nano-Glo® Luciferase Assay System (Promega) and the Tecan Infinite®M1000 plate reader.

## Measurement of itaconate by LC-MS

RAW264.7 or other cell types (~$3 \times 10^6$/ml) treated with indicated stimuli or infected with *Salmonella* were washed with 0.9% NaCl. Samples were extracted with methanol-acetonitrile-water at 2:2:1 (v/v/v). The standards, quality, and controls (QCs) were prepared in control cell extract matrix at the appropriated concentration range (1–1000 ng/ml). The cell extracted samples together with the standards and QCs were directly injected onto LCMS system for analysis. LCMS analysis was performed on a QTRAP 6500+ (AB Sciex, Singapore) coupled to an ExionLC AD Series LC system (Shimadzu, Kyoto, Japan). Analyst software (version 1.7.1, AB Sciex, Warrington, UK) was used for data acquisition and analysis. A targeted Multiple Reaction Monitoring (MRM) method was utilized to quantify the level of itaconate. An ACQUITY UPLC® BEH C18 column (1.7 µm, 2.1 × 100 mm, P.N.186002352) coupled with InfinityLab Poroshell 120 EC-CN guard column (2.7 µm, 4.6 × 5 mm, P.N.820750-927) was utilized for the liquid chromatography separation at 40 °C. The gradient started with 98% formic acid (0.05%) (A) and 2% acetonitrile (B), maintained for 1 min, and increased to 95% B in 8 min, maintained for 2 min, then increased to 2% B in 1 min and back to 95% in 1 min and maintained at 2% B for 2 min. The total run time was 15 min. QTRAP 6500+ mass spectrometer was operated using an electrospray ionization (ESI) source in negative ion mode. MRM transition monitored for itaconate was 129.0/85.1. The source settings were as follows: gas temperature of 500 °C, Curtain Gas set to 35, Ion Spray Voltage Floating set to 4500, Ion Source Gas 1 set to 50, Ion Source Gas 2 set to 50, and ion polarity set to negative. Data acquisition and analysis were performed using the SCIEX OS software (version 1.7.0, AB Sciex, Warrington, UK).

## Bacterial infections

To infect the indicated mammalian cells, *Salmonella* strains were cultured overnight at 37 °C and diluted 1:20 into LB broth containing high-concentration NaCl (0.3 M) and grown to an OD600 of 0.9. The multiplicity of infection (MOI) of *Salmonella* is 5 for *Salmonella* Typhimurium or 10 for *Salmonella* Typhi. Cells were infected for one hour in Hank's balanced salt solution (HBSS). Cells were then washed several times with cell culture medium and incubated in culture medium containing gentamycin (100 µg/ml) to kill extracellular bacteria for one hour. In order to avoid repeated cycles of infection, cells were then washed and fresh medium containing gentamycin (10 µg/ml) was added. Cells were incubated for the specified durations as indicated in the figure legends. The number of intracellular bacteria was determined by a series of dilution and plating measurable bacteria colonies on agar plates.

## Quantitative PCR

RNA from mammalian cells was extracted using TRIzol Reagent (9109, TAKARA), and first-strand cDNA was synthesized from the total RNA employing oligo-dT primers and reverse transcriptase (RR036A, TAKARA). Real-time PCR was conducted utilizing the 2×RealStar Green Fast Mixture (A301-1, Genstar) and specific primers, analyzed with CFX96 Touch Real-Time PCR Detection System. The obtained data were normalized to the GAPDH gene, and the relative transcript abundance was determined using the Ct method.

The following primers were utilized for real-time PCR:
mouse *Mct1* forward primer, 5'- TGTATGCTGGAGGTCCTATCAG -3'
mouse *Mct1* reverse primer, 5'- CCAATGCACAAGTAAAGTTCCTG -3'

mouse *Mct4* forward primer, 5'- GCCACCTCAACGCCTGCTA-3'
mouse *Mct4* reverse primer, 5'- TGTCGGGTACACCCATATCCTTA -3'

mouse *Gapdh* forward primer, 5'- GAAGGGCTCATGACCACAGT-3'
mouse *Gapdh* reverse primer, 5'- GGATGCAGGGATGATGTTCT -3'
mouse *Cd86* forward primer, 5'- TCAATGGGACTGCATATCTGCC -3'
mouse *Cd86* reverse primer, 5'- GCCAAAATACTACCAGCTCACT -3'
mouse *Cd206* forward primer, 5'- GAGGGAAGCGAGAGATTATGGA -3'
mouse *Cd206* reverse primer, 5'- GCCTGATGCCAGGTTAAAGCA -3'
mouse *Lamp1* forward primer, 5'- CAGCACTCTTTGAGGTGAAAAAC -3'
mouse *Lamp1* reverse primer, 5'- ACGATCTGAGAACCATTCGCA -3'
mouse *Mcoln1* forward primer, 5'- CTGACCCCCAATCCTGGGTAT -3'
mouse *Mcoln1* reverse primer, 5'- GGCCCGGAACTTGTCACAT -3'

mouse *Tmem55b* forward primer, 5'- TACGGAGCCGGTAAACAT GC -3'

mouse *Tmem55b* reverse primer, 5'- GGCTAGTCAAGGGTGAGTA GG -3'

mouse *Gns* forward primer, 5'- TGTGCGGCTATCAGACCTTTT -3'

mouse *Gns* reverse primer, 5'- CAGGGCATACCAGTAACTCCA -3'

mouse *Atp6v1h* forward primer, 5'- TCTGATGACGCACATCTCC AA -3'

mouse *Atp6v1h* reverse primer, 5'- GCTGACACGCTGGTGATTT TC -3'

**Western Blotting and Co-immunoprecipitation Assay**
The mammalian cells or murine tissue were lysed and extracted by NP-40/RIPA buffer. The protein samples were denatured and loaded into 8–10% SDS-polyacrylamide gel. The proteins in gel were subsequently transferred onto polyvinylidene fluoride (PVDF, BIO-RAD, 1620177) membranes, blocked with 5% (w/v) milk at room temperature for around one hour. PVDF membranes were incubated with primary antibodies against IRG1, MCT1, MCT4, GADPH, NRF2, or other indicated targets overnight at 4 °C. Following this, the membranes were further incubated with the secondary antibody, and the blots were visualized using Amersham ImageQuant 800 Western blot imaging systems.

For immunoprecipitation, whole-cell extracts were collected and subjected to be analyzed by western blotting. These extracts were then incubated overnight with anti-Flag agarose gels (A2220, from Sigma). Subsequently, the beads underwent 5–6 washes with low-salt lysis buffer, and immunoprecipitated proteins were eluted using 1 × SDS Loading Buffer and separated by SDS-PAGE. The samples were further analyzed by Western blotting.

**Microscopy**
RAW264.7 or the other indicated mammalian cells were plated on glass coverslips and incubated overnight. Cells were infected with the indicated *Salmonella* strains or treated with the indicated reagent for certain time points. The samples were washed with PBS for several times, fixed in 4% paraformaldehyde, washed with PBS, and incubated with blocking buffer (0.3% Triton-X) for around one hour. Then the samples were treated with the indicate antibody or reagent, and finally mounted on slides. Slides were imaged using an inverted confocal laser scanning microscope Zeiss LSM 900, and analyzed by Image J and ZEN Microscopy Software. The number of intracellular *Salmonella* per cell was quantified and analyzed with Opera Phenix Plus High Content Screening System.

**Flow cytometry**
BMDM were properly differentiated using the protocol described previously. For flow cytometry, BMDM were washed and resuspended in ice-cold FACS buffer (PBS supplemented with 0.5–1% BSA) at a concentration of $1–5 \times 10^6$ cells/ml. Cells were incubated with fluorophore-conjugated primary antibodies (CD11b, CD86, and CD206) for 30 min at room temperature in the dark. Following antibody staining, cells were washed thoroughly and resuspended in FACS buffer. For fixation, samples were treated with 4% paraformaldehyde and washed with FACS buffer for 3 times prior to analysis. Data were acquired on a flow cytometer (Attune Nxt, Thermo) on the day of staining or after short-term storage at 4 °C in fixative. M1 macrophages were identified as CD11b[+]CD86[+], and M2 macrophages were analyzed based on CD11b[+]CD206[+] populations.

**pH modulation**
The pH of Dulbecco's Modified Eagle Medium (DMEM) or RPMI 1640 Medium (Roswell Park Memorial Institute 1640 Medium) was adjusted using H+/OH- buffers and measured by a pH meter. To alter extracellular pH, cells were incubated with culture media that had been adjusted to the desired pH. To modulate intracellular pH, cells were incubated with culture medium adjusted to the desired pH, supplemented with 30 mM $NH_4Cl$ and 10 µM EIPA (5-(N-ethyl-N-isopropyl)-Amiloride) for 15 min. Following this, the intracellular pH was stabilized by re-incubating the cells in the pH-adjusted medium containing EIPA (10 µM) for the specified duration[26].

**LC-MS-based untargeted metabolomics**
Wild-type (WT) or *Mct4*[−/−] BMDMs ($3 \times 10^6$ cells per sample) were treated with AZD3965 as indicated in the Figure legends. Samples were extracted with methanol-acetonitrile-water at 2:2:1 (v/v/v). Extracts of these cells were dried in a vacuum concentrator at 4 °C and analyzed using liquid chromatography-mass spectrometry (LC-MS). Data acquisition was conducted with a UHPLC system (Vanquish, Thermo Scientific) coupled to an Orbitrap mass spectrometer (Exploris 480, Thermo Scientific). LC separation utilized a Waters BEH amide column and a Phenomenex Kinetex C18 column. Mobile phases, linear gradient elution, and ESI source parameters followed previously published methods[62]. Metabolite annotation was performed using MetDNA (http://metdna.zhulab.cn/)[63,64].

**Statistics**
Data were analyzed by GraphPad Prism 9.0 software. Unless otherwise indicated, statistical significance was determined by two-sided Student's *t*-test or ANOVA.

**Reporting summary**
Further information on research design is available in the Nature Portfolio Reporting Summary linked to this article.

## Data availability
All related data are available in the main text, supplementary materials, and auxiliary files. More information and data that support the findings of this study are available from the corresponding author upon reasonable request. The raw data files of the metabolomics data generated in this study have been deposited and are available in MassIVE repository (accession:MSV000099371;https://massive.ucsd.edu/ProteoSAFe/dataset.jsp?accession=MSV000099371) or the National Omics Data Encyclopedia (accession: OEP00006594). Source data are provided with this paper.

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

## Acknowledgements

We thank Professor Feng Shao (National Institute of Biological Sciences), Professor Gong Cheng (Tsinghua University), Professor Xun Sun (West China School of Pharmacy, Sichuan University), Professor Yue Xu (Shanghai Jiao Tong University), Professor Qiankun Wang (Shenzhen Bay Laboratory), Professor Min Zheng (Shenzhen Bay Laboratory), Professor Qing Peng (Shenzhen Bay Laboratory) for kindly providing us with the related bacterial strains, mammalian cell lines or other important experimental materials. We thank Professor Lei Zhou (Shenzhen Bay Laboratory) for providing us with *Xenopus laevis* and Professor Lang Rao and Dr. Lei Cao (Shenzhen Bay Laboratory) for providing us with detailed experience in generating liposomes. We thank Professor Zhengjiang Zhu (Shanghai Institute of Organic Chemistry, Chinese Academy of Sciences) for providing technical support in detecting itaconate and other metabolites. We thank Professor Lin Wu, Bo Zhang, Xinhai Chen, Hao Xu, and others for their valuable suggestions to this study during routine faculty lunches at Shenzhen Bay Laboratory. We acknowledge grants support from the National Natural Science Foundation of China (82302541, M.C.), the Shenzhen Medical Research Fund (A2303045, M.C.), the Shenzhen Bay Scholar Fellowship (S229100002, M.C.) and Shenzhen Bay Laboratory Start-up Funds (21330041, M.C.). We are also grateful for the support from the core facilities, including Biological Imaging platform (with help from Dr. Shan Liu), Multi-omics mass spectrometry platform (with help from Dr. E Li), the Biosafety Level 2 (BSL-2) and Animal Biosafety Level 2 (ABSL-2) Laboratories of the Institute of Infectious Diseases, Shenzhen Bay Laboratory, for their support and assistance during this research. We would like to thank Ying Chen from Pellucid Optics Technology (Nantong) Co., Ltd. for capturing the ODT images.

## Author contributions

Conceptualization: M.C. Methodology: Q.M., C.L., Y.C., M.C., X.W. and F.L. Investigation: Q.M., C.L., M.C., Y.Z., Y.C., X.C. and B.Z. Visualization: Q.M., C.L. and M.C. Funding acquisition: M.C. Project administration: M.C. and Q.M. Supervision: M.C. Writing-original draft: M.C. Writing-review and editing: M.C. and Q.M.

## Competing interests

The authors declare no competing interests.
