## [Transparent Peer Review file · Nature Communications]

Itaconate transport across the plasma membrane and Salmonella-containing vacuole via MCT1/4 modulates macrophage antibacterial activity

Corresponding Author: Professor Meixin Chen

Version 0:

Reviewer comments:

Reviewer #1

(Remarks to the Author)

The authors have addressed my concerns.

Reviewer #4

(Remarks to the Author)

The revised manuscript from Meng et al. made a significant effort in responding to the comments of reviewer 3. Overall, the comments have been addressed but some points need to be clarified or further assessed.

1/ For R#3, question 1: Authors now show that the human transporters MCT1 and MCT4 can directly mediate itaconate uptake using expression of recombinant MCT1/4 in *Xenopus laevis* oocytes as well as in liposomes (new Figure 5). However, for experiment with reconstructed MCT1/MCT4 liposomes (figure 5f, 5g, 5h), how authors can manage the orientation of the transporters (in/out), and consequently determine whether they will assess uptake or export of itaconate ?

2/ For R#3, question 2: Authors now assessed lactate and itaconate transport in 293T and U2OS cells overexpression WT or dead mutants of MCT1/MCT4 in a new Figure 4j-n and extended Figure 6f-i. Regarding itaconate transport, they showed that inactive mutants of MCT1/MCT4 inhibited itaconate uptake by the cells (Figure 4j-n). However, regarding lactate transport, they concluded that WT MCT1/MCT4 enhanced lactate secretion compared to mutants (line 238-239), but authors rather showed that inactive mutants of MCT1/MCT4 increased intracellular lactate compared to WT (extended Figure 6f-i). To conclude regarding lactate secretion, authors should measure and show extracellular lactate levels. Indeed, decrease intracellular levels does not necessarily means increased export, as lactate could be metabolized (conversion into pyruvate by lactate deshydrogenase).

3/ For R#3, question 3: To exclude a secondary pathway involved, authors validated their conclusions into new models of transport, *Xenopus* oocytes and proteoliposomes, with medium containing only itaconate (and not other metabolites sensed by MCT1/MCT4). This demonstration is valid regarding itaconate transport. Regarding the impact of bacterial growth, the effect could be not direct, but indirect through modulation of immune response that subsequently impact bacterial growth. Blocking MCT1/MCT4 could affect the functionality of macrophage, that in turn inhibit bacterial growth. See point 6/ below for further suggestion to address this potential secondary pathway.

4/ I think that the title is not sustained by the experimental work. The title states that "the transport of itaconate [...] via MCT1/4 modulates antibacterial immune responses". In this work, authors just show an effect on inhibition of bacterial growth, but not at all on antibacterial immune responses. This point needs to be further addressed, by investigating for example the lysosomal biogenesis, bacteriolysis activity of macrophages, or other functional features related to antibacterial immune responses in their in vivo model.

5/ This work has been entirely done with tumoral cell lines [RAW264 (murine macrophage line), HeLa (human cervical cancer line), HEK293T (adenovirus-immortalized human embryonic kidney cells), DC2.4 (mouse DC cell line)] or murine BMDC or BMDM cells. MCT1/MCT4 expression levels, sensibility to inhibitor, as well as ability to sense metabolites could be dependent on cell type and species. It is well documented that MCT1/MCT4 expression levels and affinity for metabolites are different on tumor cells compared to immune cells. This study could gain interest if authors perform some key findings in vitro with human primary macrophages, or in vivo with humanized mice (bearing a human immune system), as the finality is to develop therapeutic strategies in humans targeting itaconate/MCT pathway, able to reinvigorate antibacterial immunity. Therefore, the use of human primary immune cells is crucial to draw such conclusion.

6/ Another key point is to assess how itaconate / MCT inhibition affect macrophage M1/M2 polarization, toward pro-inflammatory or anti-inflammatory state. Itaconate has been shown to be rather anti-inflammatory (<https://doi.org/10.1016/j.cmet.2025.03.004>). Authors should investigate the impact of MCT1/MCT4 blocking and/or itaconate level on the immunomodulatory role of macrophage, at phenotypic and functional levels. For example, investigate some surface markers (such as CD86, CD206) or cytokine secretion (such as TNF- α , IL-1 β , IL-6, MCP-1, IL-10, TGF- β), or other molecules (arginase 1) that could distinguish M1 versus M2 polarization of macrophages. It will allow to link the inhibition of bacterial growth with some markers of immune response.

7/ It would be interesting to decipher differences between MCT1 and MCT4 regarding itaconate transport. How do they compare regarding uptake / export, threshold for sensing itaconate, and competition for lactate transport ?

8/ Line 66, it is mentioned "anti-bacterial infection". This formulation is weird. It should be either "anti-bacterial defense" or "bacterial infection".

Version 1:

Reviewer comments:

Reviewer #4

(Remarks to the Author)

The revised manuscript from Meng et al. addressed most of the concerns. They provide explanation regarding the orientation of the transporters, further addressed lactate transport by MCT1/4, provided data with primary human macrophages, and more deeply discussed differences between MCT1 and MCT4.

There are two remaining concerns:

1/ Regarding the modulation of immune response that could subsequently impact bacterial growth, I think the investigations are not sufficient. The authors should assess whether blocking MCT1/MCT4 could affect the functionality of macrophage, such as cytokine secretion profile or capacity of phagocytosis. These data need to be added in the paper. We need to clearly know if the effect of MCT1/MCT4 blocking is direct (direct impact on bacterial growth) or indirect through modulation of the functionality of macrophages, or both.

To assess M1/M2 macrophages polarization, authors performed CD86 and CD206 surface staining on BMDMs treated with Su3118 and infected with bacteria. We don't know the timing, and in which population expression is defined (% within which cell population ?). To really assess how itaconate / MCT inhibition affect macrophage M1/M2 polarization toward pro-inflammatory or anti-inflammatory state, authors should investigate the impact of MCT1/MCT4 blocking and/or itaconate level on the immunomodulatory role of macrophage, at functional levels: investigate cytokine secretion (such as TNF- α , IL-1 β , IL-6, MCP-1, IL-10, TGF- β), or other molecules (arginase 1) that could distinguish M1 versus M2 polarization of macrophages. They should use positive control of polarization, such as culture with LPS and IL4. Functionality of macrophages could also be assessed through investigation of lysosomal biogenesis, and bacteriolysis activity of macrophages.

2/ The new title is still ambiguous. Authors changed "antibacterial immune responses" by "antibacterial efficacy". Indeed, authors showed an effect on inhibition of bacterial growth, but not at all on antibacterial immune responses, or on the efficacy of macrophages to clear infection. I think it worth assessing some functionality of macrophages to really conclude on this point and keep the proposed title.

Version 2:

Reviewer comments:

Reviewer #4

(Remarks to the Author)

The authors have addressed most of the concerns. Here are my remaining comments:

1/ I'm not convinced by the gating strategy to identify M1/M2 macrophages (Suppl Figure 2k). Authors gated the BMDMs as

CD11b+ CD86+ and CD11b+ CD206+ respectively, but they should gate CD11b+ CD86+CD206- and CD11b+ CD206+CD86-. Indeed, with the current gating strategy, there is 54% of M1 and 82% of M2, so a total largely superior to 100%, meaning that a large majority of cells co-express both markers, and are not clearly M1 or M2. A dot plot pre-gated on CD11b+ and displaying CD86 versus CD206 should help depicting more clearly M1 and M2 macrophages on the same dot plot.

In contrast, the quantification of mRNA levels (Suppl Figure 2I) clearly shows that LPS induces CD86 expression (marker of M1), and IL4 triggers CD206 (marker of M2), as expected. In addition, *S. Typhi* induced CD86 (4-fold induction) but not CD206, sustaining a M1 polarization.

The authors should clarify this point, especially the gating strategy for M1/M2 and divergence between flow cytometry and mRNA analyses.

2/ Regarding the title:

Authors shows that transport of itaconate is facilitated by RAB32, yet this component is not specific to immune cells. The NRF2 pathway, induced by itaconate, even if it is involved in antibacterial defense mechanisms, plays also an important role in the regulation of genes that control the expression of proteins critical in the detoxication and elimination of ROS. These two pathways are not specific of immune responses. As the new data demonstrate no effect of itaconate on macrophage polarization, functionality, capacity of phagocytosis or lysosomal biogenesis (which would be pertinent immune response elements), the link between itaconate and antibacterial immune responses is really not evident and not sustain by the content of the paper. A suggestion would be: "Transport of itaconate [...] modulates macrophage' antibacterial activity".

We sincerely thank the Editors and Reviewers for their positive feedback and valuable suggestions. We have carefully and thoroughly considered the suggestions and concerns raised by the Reviewer#4. We believe we have addressed all comments thoroughly and really hope the Editors and Reviewer#4 will be pleased with our revised manuscript. To facilitate the review process, we have showed all changes in the manuscript text file with **color highlighting**. Thank you so much for your understanding.

Response to Reviewers' Comments

Reviewer #1 (Remarks to the Author):

The authors have addressed my concerns.

We sincerely thank Reviewer #1 and #2 for the support of the publication.

Reviewer #4 (Remarks to the Author):

The revised manuscript from Meng et al. made a significant effort in responding to the comments of reviewer 3. Overall, the comments have been addressed but some points need to be clarified or further assessed.

#1. *For R#3, question 1: Authors now show that the human transporters MCT1 and MCT4 can directly mediate itaconate uptake using expression of recombinant MCT1/4 in *Xenopus laevis* oocytes as well as in liposomes (new Figure 5). However, for experiment with reconstructed MCT1/MCT4 liposomes (figure 5f, 5g, 5h), how authors can manage the orientation of the transporters (in/out), and consequently determine whether they will assess uptake or export of itaconate?*

To Reviewer# Figure 1. MCT1/4 transports itaconate into liposomes.

Answer: We appreciate the Reviewer's valuable comments. Physiologically, many SLC proteins are highly dynamic (Cell 2021, 184(2): 370–383), which is challenging to fully recapitulate *in vitro*. To make our results reliable, we performed the liposome-based transport assay by following well-established protocols, as described in the manuscript (Cell, 2021, 184(2): 370-383; Cell Metabolism, 36(3), 498-510; Nature protocols, 3(2), 256-266).

Importantly, two key points assure that itaconate uptake was measured in our assay. **First, the transport direction of MCT1/4 is mainly determined by the concentration gradient of substrates** (Cell, 2021, 184(2): 370-383; Physiological Reports, 2022, 10(17)). The MCT1/4 liposomes contain no itaconate inside. When liposomes were incubated in itaconate-containing buffer, the concentration of itaconate is much higher out of liposomes. MCT1/4 is prone to transport itaconate into liposomes (**To Reviewer# Figure 1**). **Second, only itaconate in the liposomes were collected.** We washed the MCT1/4 liposomes after the incubation with itaconate. Only itaconate in the liposomes was extracted and analyzed. Besides, we took advantage of a *Xenopus* oocytes-dependent assay and other experiments (Figure 3 and 4 in the manuscript) to examine itaconate transport, which support MCT1 and MCT4 transport itaconate.

#2. For R#3, question 2: Authors now assessed lactate and itaconate transport in 293T and U2OS cells overexpression WT or dead mutants of MCT1/MCT4 in a new Figure 4j-n and extended Figure 6f-i.

Regarding itaconate transport, they showed that inactive mutants of MCT1/MCT4 inhibited itaconate uptake by the cells (Figure 4j-n). However, regarding lactate transport, they concluded that WT MCT1/MCT4 enhanced lactate secretion compared to mutants (line 238-239), but authors rather showed that inactive mutants of MCT1/MCT4 increased intracellular lactate compared to WT (extended Figure 6f-i). To conclude regarding lactate secretion, authors should measure and show extracellular lactate levels. Indeed, decrease intracellular levels does not necessarily means increased export, as lactate could be metabolized (conversion into pyruvate by lactate dehydrogenase).

To Reviewer# Figure 2. (A, B) U2OS cells were transfected with MCT1 (A), MCT4 (B) or their mutants. 24 h later, the media was replaced with fresh media, and extracellular lactate levels were measured at 1, 3, and 6 h using the Lactate-Glo™ Assay.

Answer: As suggested by the Reviewer, we examined the extracellular lactate level. Compared to the cells transfected with wild-type MCT1/MCT4, the cells transfected with the mutants show reduced lactate export (**To Reviewer# Figure 2A, 2B**, refer to new Supplementary Data Fig. 6j, 6k). These findings along with the results of Supplementary Figure 6h, 6i in the manuscript support that MCT1/4 mutations attenuated lactate transport.

#3. For R#3, question 3: To exclude a secondary pathway involved, authors validated their conclusions into new models of transport, *Xenopus oocytes* and proteoliposomes, with medium containing only itaconate (and not other metabolites sensed by MCT1/MCT4). This demonstration is valid regarding itaconate transport. Regarding

the impact of bacterial growth, the effect could be not direct, but indirect through modulation of immune response that subsequently impact bacterial growth. Blocking MCT1/MCT4 could affect the functionality of macrophage, that in turn inhibit bacterial growth. See point 6/ below for further suggestion to address this potential secondary pathway. #6. Another key point is to assess how itaconate / MCT inhibition affect macrophage **M1/M2 polarization**, toward pro-inflammatory or anti-inflammatory state. Itaconate has been shown to be rather anti-inflammatory (<https://doi.org/10.1016/j.cmet.2025.03.004>). Authors should investigate the impact of MCT1/MCT4 blocking and/or itaconate level on the immunomodulatory role of macrophage, at phenotypic and functional levels. For example, investigate some surface markers (such as CD86, CD206) or cytokine secretion (such as TNF- α , IL-1 β , IL-6, MCP-1, IL-10, TGF- β), or other molecules (arginase 1) that could distinguish M1 versus M2 polarization of macrophages. It will allow to link the inhibition of bacterial growth with some markers of immune response.

To Reviewer# Figure 3. (A, B) BMDMs (bone marrow-derived macrophages) were treated with DMSO/Su3118 (5 μ M) and infected with or without *Salmonella* Typhi (24 h). The cells were stained with anti-CD11b, anti-CD86 and anti-CD206. The expression of M1 surface marker CD86 or M2 surface markers CD206 of cells were measured by Flow cytometry.

Answer: We sincerely appreciate these suggestions from the Reviewer. In our study, we have examined the antibacterial role of MCT1 and MCT4 inhibition in bacteria-infected macrophages. On the basis of this, to assess whether the antibacterial effect involves MCT1/4-mediated macrophage polarization, we analyzed macrophage polarization during bacterial infection.

Flow cytometry results confirmed that inhibiting MCT1 and MCT4 by Su3118 did not significantly alter M1/M2 marker protein levels in bacteria-infected BMDMs (**To Reviewer# Figure 3A, 3B**). These results rule out an effect of MCT1 and MCT4 on macrophage polarization during bacterial infection.

#4. I think that the title is not sustained by the experimental work. The title states that “the transport of itaconate [...] via MCT1/4 modulates antibacterial immune responses”. In this work, authors just show an effect on inhibition of bacterial growth, but not at all on antibacterial immune responses. This point needs to be further addressed, by investigating for example the lysosomal biogenesis, bacteriolysis activity of macrophages, or other functional features related to antibacterial immune responses in their *in vivo* model.

Answer: We thank the Reviewer for the helpful reminder. To address the Reviewer’s concern, the title of our article has been modified as “MCT1/4-dependent itaconate transport across the plasma membrane and *Salmonella*-containing vacuole modulates antibacterial efficacy”.

The previous one : *Transport of itaconate across the plasma membrane and Salmonella-containing vacuole via MCT1/4 modulates antibacterial immune responses.*

#5. This work has been entirely done with tumoral cell lines [RAW264 (murine macrophage line), HeLa (human cervical cancer line), HEK293T (adenovirus-immortalized human embryonic kidney cells), DC2.4 (mouse DC cell line)] or murine BMDC or BMDM cells. MCT1/MCT4 expression levels, sensibility to inhibitor, as well as ability to sense metabolites could be dependent on cell type and species. It is well documented that MCT1/MCT4 expression levels and affinity for metabolites are different on tumor cells compared to immune cells. This study could gain interest if authors perform some key findings *in vitro* with human primary macrophages, or *in vivo* with humanized mice (bearing a human immune system), as the finality is to develop therapeutic strategies in humans targeting itaconate/MCT pathway, able to reinvigorate antibacterial immunity. Therefore, the use of human primary immune cells is crucial to draw such conclusion.

To Reviewer# Figure 4. (A) Human monocyte-derived macrophages (hMDM) treated with DMSO/Su3118 (5 μM) were infected with *Salmonella* Typhi for 22 h. The number of bacteria in cells was

examined. **(B, C)** The level of itaconate from hMDM treated with Su3118 (5 μ M) and infected with *Salmonella* Typhi for around 22 hours (LC-MS). **(D)** Intracellular itaconate from hMDM treated with Su3118 (5 μ M), and infected by *Salmonella* Typhi for 22 h was determined by itaconate biosensor. The levels of luciferase were measured.

Answer: We agree with the Reviewer that it's necessary to repeat key experiments using human monocyte-derived macrophages (hMDM), obtained as described in the methods in the manuscript. With the limited primary human cells, we then have tried to examine itaconate transport and the related antibacterial effect.

- (1) **CFU assay:** A lower bacterial load was detected in hMDM cells with MCT1 and MCT4 impairments **(To Reviewer# Figure 4A, refer to Fig. 1h in the new manuscript)**.
- (2) **LC-MS analysis:** The inhibition of MCT1 and MCT4 by Su3118 impaired the release of itaconate **(To Reviewer# Figure 4B, refer to Fig. 3g, 3h in the new manuscript)**. Spontaneously, this inhibition led to increasing itaconate intracellularly in hMDM **(To Reviewer# Figure 4C)**.
- (3) **Itaconate-biosensor assay:** We found that more bacteria were targeted by itaconate in hMDM when MCT1 and MCT4 activity was suppressed, detected by using itaconate biosensor **(To Reviewer# Figure 4D, refer to Fig. 2e in the new manuscript)**. These finding implied that itaconate exerted its antibacterial effect in a MCT1/4-dependent manner **(To Reviewer# Figure 4A)**.

These findings are consistent with our previous observations in primary macrophages from mice or RAW264.7 cells, which support our important conclusions.

#7. *It would be interesting to decipher differences between MCT1 and MCT4 regarding itaconate transport. How do they compare regarding uptake / export, threshold for sensing itaconate, and competition for lactate transport?*

Answer: We highly agree with the Reviewer and the Editors that it would be interesting to learn more about the different roles of MCT1 and MCT4 in transporting itaconate. As kindly noted by the Editors, we have expanded the relevant discussion in the revised manuscript. We hope this revised discussion satisfactorily acknowledges the importance of the Reviewer's suggestion.

Line 373 to 387 in the revised manuscript:

Additionally, MCT1 is a widely expressed protein, found in the macrophages as well as various mouse tissues such as spleen, liver, brain, and kidney, while MCT4 is an inducible protein in the macrophages, spleen and liver, but not expressed in the brain and kidney **(Fig. 1o, 1p, and Supplementary Fig. 1o)**. These observations suggest that MCT1 and MCT4 may function differently in the uptake of itaconate across different tissues. Given that both transporters are also known to carry lactate or other related substrates, their efficiency and specificity for itaconate transport may be

influenced by substrate competition, depending on the cellular context and metabolic demands. Their transport activity is probably not fixed but dynamically regulated by the availability of competing molecules. We found that the presence of itaconate attenuated lactate uptake (**Figure 4h**). And lactate also reduced itaconate uptake (**Figure 4i**). Since MCT1 has a higher affinity for lactate compared to MCT4 (Cell reports, 2018, 25(11): 3047-3058), it is possible that lactate competition has a greater impact on itaconate transport through MCT1. Further investigations along these lines would be valuable in the future.

#8. *Line 66, it is mentioned “anti-bacterial infection”. This formulation is weird. It should be either “anti-bacterial defense” or “bacterial infection”.*

Answer: We are really sorry for this mistake. We have corrected the mentioned words (Line 64) as suggested by the Reviewer and highlighted them in yellow in the revised manuscript.

Response to Reviewers' Comments

Once again, we extend **our sincere thanks** to Reviewer #4 for the valuable comments that have helped us further improve the manuscript. We have thoroughly considered the suggestions raised by Reviewer#4 and tried our best to address the concerns. A total of approximately **32 new panels**, have been added to clearly illustrate our findings in the manuscript or the response letter. To facilitate the review, we have marked the changes in the manuscript with color highlighting.

Reviewer #4 (Remarks to the Author):

The revised manuscript from Meng et al. addressed most of the concerns. They provide explanation regarding the orientation of the transporters, further addressed lactate transport by MCT1/4, provided data with primary human macrophages, and more deeply discussed differences between MCT1 and MCT4.

We deeply thank the reviewer for the positive feedback regarding our efforts in the previous version of the manuscript.

There are two remaining concerns:

1# *Regarding the modulation of immune response that could subsequently impact bacterial growth, I think the investigations are not sufficient. The authors should assess whether blocking MCT1/MCT4 could affect the functionality of macrophage, such as cytokine secretion profile or capacity of phagocytosis. These data need to be added in the paper. We need to clearly know if the effect of MCT1/MCT4 blocking is direct (direct impact on bacterial growth) or indirect through modulation of the functionality of macrophages, or both.*

To assess M1/M2 macrophages polarization, authors performed CD86 and CD206 surface staining on BMDMs treated with Su3118 and infected with bacteria. We don't know the timing, and in which population expression is defined (% within which cell population?). To really assess how itaconate/MCT inhibition affect macrophage M1/M2 polarization toward pro-inflammatory or anti-inflammatory state, authors should investigate the impact of MCT1/MCT4 blocking and/or itaconate level on the immunomodulatory role of macrophage, at functional levels: investigate cytokine secretion (such as TNF- α , IL-1 β , IL-6, MCP-1, IL-10, TGF- β), or other molecules (arginase 1) that could distinguish M1 versus M2 polarization of macrophages. They should use positive control of polarization, such as culture with LPS and IL4. Functionality of macrophages could also be assessed through investigation of lysosomal biogenesis, and bacteriolysis activity of macrophages.

Answer: We appreciate these detailed suggestions from the Reviewer. We have carefully examined the effect of MCT1 and MCT4 blocking on the functionality of macrophages. The newly generated data have been incorporated into the manuscript.

Topic one: polarization and cytokine expression

(1) Flow Cytometry Gating Strategy

Supplementary Fig. 2k: Representative flow cytometry plots of CD11b⁺ BMDMs showing CD86 and CD206 expression following *Salmonella* (*S. Typhi*) infection (24 h) with or without Su3118 (5 μM, 22h) treatment.

In our previous response, we stated that “Su3118 did not alter M1/M2 marker protein levels in bacteria-infected BMDMs.” We apologize for not including the gating strategy at that time, which may have caused confusion. As clarified here, BMDMs were properly differentiated using the protocol described previously. Cells were stained with antibodies against CD11b, CD86, and CD206. M1 macrophages were identified as CD11b⁺CD86⁺. M2 macrophages were analyzed based on CD11b⁺CD206⁺ populations (**Supplementary Fig. 2k**).

The detailed gating strategy has now been included in this letter for clarity, and the full method has also been added to the "METHODS" section of the revised manuscript. All flow cytometry data throughout the study were acquired and analyzed following this consistent approach.

(2) Blocking MCT1 and MCT4 does not significantly change the polarization of macrophages.

Fig. 2m: Flow cytometry data from Supplementary Figure 2k showing the percentages of CD11b⁺CD86⁺ and CD11b⁺CD206⁺ BMDMs. **Fig. 2n:** Flow cytometry data from BMDMs treated with Su3118 (5 μ M, 24 h), LPS (100 ng/ml, 24 h), or IL-4 (20 ng/ml, 24 h), showing the percentages of CD11b⁺CD86⁺ and CD11b⁺CD206⁺ cells. **Supplementary fig. 2l:** Relative mRNA levels of the indicated cytokines in BMDMs treated with Su3118 (5 μ M, 22 h) with LPS (100 ng/ml, 18 h)/IL-4 (20ng/ml, 18 h)/S. Typhi infection (24 h) were determined by quantitative real-time PCR.

To assess whether the antibacterial effect involves MCT1/4-mediated macrophage polarization, we analyzed repeatedly macrophage polarization during bacterial infection or under LPS/IL4 stimulation. Blocking MCT1 and MCT4 had minimal impact on the polarization of macrophages under during bacterial infection (**Fig. 2m**). Representative flow plots from BMDMs were shown (**Supplementary Fig. 2k**). Furthermore, the polarization of the majority of macrophages under LPS/IL4 stimulation were not influenced by the inhibition of MCT1 and MCT4 (**Fig. 2n**). Su3118 also did not significantly change the mRNA level of *Cd206* or *Cd86* in BMDMs in response to bacterial infection or LPS/IL4 treatment (**Supplementary Fig. 2l**).

These data rule out the possibility that blocking MCT1 and MCT4 reduces bacterial survival by altering the polarization of the majority of macrophages. Therefore, our conclusion remains consistent with the previous one: blocking MCT1 and MCT4 did not significantly alter macrophage polarization.

R-Fig. 1: Relative mRNA levels of the indicated cytokines in RAW264.7 cells treated with Su3118 (5 μ M, 24h) with LPS (100 ng/ml, 18 h)/IL4 (20ng/ml, 18h) were determined by quantitative real-time PCR. Expression was normalized to untreated cells.

Regarding cytokine functionality, we have further analyzed the expression of cytokines and markers that differentiate M1 from M2 macrophage polarization. We found that inhibiting MCT1 and MCT4 in macrophages did not remarkably influence the mRNA level of cytokines, including *Arg1*, *Tgf- β 1*, *Mrc1*, *Mcp1*, *Irf5*, *Tnf α* , *Il6*, *Il1 β* , *Il-10*, *Cd86*, *Cd80* and *Cd206* (**R-Fig. 1**). These data suggest that the inhibition of MCT1 and MCT4 neither changes M1/M2 polarization nor alters the expression of pro- or anti-inflammatory cytokines or markers in macrophages. Itaconate accumulation caused by Su3118 may not be sufficient on its own and may require co-stimulatory signals to modulate these cytokines. It's also possible that the accumulation of other substrates of MCT1 and MCT4 in cells caused by Su3118 is also involved in these results.

Fig. 4e

Fig. 4f

Fig. 4e, f: RAW264.7 (e) or HeLa (f) cells were treated with MG132 (10 μ M), itaconate (1 mM) or Su3118 (10 μ M) as indicated. The cell lysates were analyzed by the immunoblotting with anti-NRF2 and anti-GAPDH.

We did observe that less itaconate uptake caused by inhibiting MCT1 and MCT4 transport activity attenuated the effect of itaconate on an oxidative-stress regulator NRF2 (**Fig. 4e, f**). NRF2 has been reported to be upregulated by itaconate and has a positive role in resisting bacterial infection (Cell Reports 37.3 (2021)). This implies that

itaconate imported by MCT1 and MCT4 also participates in NRF2-dependent antibacterial immunity in cells. MCT1 and MCT4 probably affect certain immune pathways (e.g. NRF2-related pathway) in an itaconate-dependent manner, which in turn has a role in regulating bacterial survival intracellularly.

Topic two: phagocytosis and bacteriolysis

(1) Blocking MCT1 and MCT4 does not affect macrophage phagocytosis.

Supplementary Fig. 1e, g, h: RAW264.7 cells (**e**) pretreated with Su3118 (0, 10 μ M, 22 h) were infected with *S. Typhi*. The number of bacteria in cells was determined at 0.5 h after infection. (**g, h**) RAW264.7 or BMDMs were pretreated with Su3118 (0 or 10 μ M, 22 h) and infected with mScarlet-expressing *S. Typhi*. At the indicated time, cells were fixed, stained with Hoechst, and imaged, quantified using the Opera Phenix Plus system.

To evaluate whether blocking MCT1/MCT4 affects the phagocytic capacity of macrophages, we conducted experiments assessing their short-term uptake of *Salmonella* (within one hour of infection).

A colony-forming unit (CFU) assay was used to assess the bacterial uptake 0.5 h post-infection, which reflects the phagocytosis of macrophages. Even with long-term inhibition of MCT1 and MCT4 transport activity via prior treatment with Su3118, it did not alter bacterial uptake in macrophages (**Supplementary Fig. 1e**).

Additionally, a fluorescence-based bacterial uptake assay using a high-content imaging system was performed to evaluate phagocytosis in both RAW264.7 and primary macrophages. The results were consistent with the CFU assay, showing that Su3118 did not affect *Salmonella* uptake at either 30 minutes or 1 h post-infection (**Supplementary Fig. 1g, h**).

(2) Blocking MCT1 and MCT4 does not affect lysosomal biogenesis.

Supplementary Fig. 2i and Fig. 2l: Relative mRNA levels of lysosomal genes in BMDMs treated with Su3118 (5 μ M, 24 h), with or without LPS (100 ng/ml, the indicated times), were measured by RT-PCR. Expression was normalized to untreated cells, and triplicate values were shown as heatmaps. **Supplementary Fig. 2j:** RAW264.7 cells treated with LPS (100 ng/ml, 24 h) or IL-4 (20 ng/ml, 24 h). The cell lysates were analyzed by immunoblotting with anti-Lamp1 and anti- β -actin.

The reviewer also provided suggestions about the investigation of lysosomal biogenesis of macrophages. We appreciate the suggestion from the Reviewer. Lysosomal biogenesis is essential for macrophage bacteriolytic activity. We found LPS activated five key genes of lysosomal biogenesis in a time-dependent manner (**Supplementary Fig. 2i**), which was the same as described before (Molecular Cell 82.15 (2022): 2844-2857). When treating cells with Su3118, the MCT1 and MCT4 inhibitor, it hardly regulated the expression of the lysosomal genes (**Fig. 2l**). Su3118 also did not affect the protein level of Lamp1 (**Supplementary Fig. 2j**). Itaconate alone, as induced by Su3118, may be limited to drive lysosomal biogenesis without complementary signals. And, besides itaconate, other substrates of MCT1 and MCT4, which accumulate intracellularly under Su3118 treatment, probably also contribute to this outcome, which requires further study. Overall, blocking MCT1 and MCT4 probably did not boost the expression of lysosomal genes.

In conclusion, our new findings above excluded that Su3118 suppresses intracellular bacterial survival through changing macrophage polarization, phagocytic activity or lysosomal biogenesis. **Importantly**, using an itaconate biosensor to detect endogenous itaconate in bacteria-containing vacuole, we have clearly confirmed that the ratio of itaconate-targeting bacteria *in vitro* or *in vivo* were regulated by MCT1 and MCT4 (**Fig. 2d-g**), which led to a substantial decrease in bacterial load upon Su3118 treatment, dependent on itaconate, observed both *in vitro* and *in vivo* (**Fig. 1k and 1n**). In general, we believe the primary effect of blocking MCT1/MCT4 is a direct impact on bacterial growth, particularly considering the influence of itaconate transport across

the bacteria-containing vacuole membrane. The use of itaconate biosensor provides compelling evidence for this direct targeting. We fully agree with the reviewer's insightful point that itaconate may impact multiple pathways (e.g. NRF2 pathway) associated with bacterial growth. This reinforces the importance of exploring these regulatory mechanisms in future research.

2# *The new title is still ambiguous. Authors changed “antibacterial immune responses” by “antibacterial efficacy”.*

Indeed, authors showed an effect on inhibition of bacterial growth, but not at all on antibacterial immune responses, or on the efficacy of macrophages to clear infection. I think it worth assessing some functionality of macrophages to really conclude on this point and keep the proposed title.

Answer: We sincerely appreciate the reviewer's thoughtful reminder. To address the reviewer's concern, we have assessed the functionality of macrophages, including macrophage polarization, cytokine profile, capacity of phagocytosis, or lysosomal biogenesis. Our findings suggest that the accumulation of itaconate caused by the inhibition of MCT1 and MCT4 probably indirectly suppresses their replication by modulating certain immune responses (e.g. NRF2 pathway) in macrophages.

The related discussion was highlighted in the manuscript (**Line 374-386**): ...The attenuation of the itaconate-related effect was shown when inhibiting its uptake via MCT1 and MCT4, as evidenced by the downregulation of NRF2 expression (Fig. 4e, f). Itaconate, imported through MCT1 and MCT4, may act as an important messenger to activate antibacterial activity or other immune responses intracellularly. Additional research is necessary to comprehensively understand the physiological effect of itaconate transported by MCT1 and MCT4.

Moreover, our findings support that MCT1 and MCT4-dependent transport of itaconate into bacteria-containing vacuole is facilitated by RAB32 (**Fig. 7a-g**). The RAB32-mediated cell-autonomous antibacterial defense represents an innate immune mechanism that has been well established in multiple previous studies (Science 369.6502 (2020): 450-455; Science 338.6109 (2012): 960-963; Cell host & microbe 19.2 (2016): 216-226; iscience 24.1 (2021)). Our data further support the role of MCT1 and MCT4 in transporting the antimicrobial metabolite itaconate during the RAB32-mediated immune response, establishing a link between metabolite transport and antimicrobial immunity.

Based on these points, our findings support our manuscript title. And, as suggested by the reviewer, we have kept the proposed title “*Transport of itaconate across the plasma membrane and Salmonella-containing vacuole via MCT1/4 modulates antibacterial immune responses*”.

Response to Reviewers' Comments

We extend our sincere thanks to Reviewer #4 for the valuable comments that have helped us further improve the manuscript. We have thoroughly considered the suggestions raised by Reviewer#4 and tried our best to address the concerns.

Reviewer #4 (Remarks to the Author):

1/ I'm not convinced by the gating strategy to identify M1/M2 macrophages (Suppl Figure 2k). Authors gated the BMDMs as CD11b+ CD86+ and CD11b+ CD206+ respectively, but they should gate CD11b+ CD86+CD206- and CD11b+ CD206+CD86-. Indeed, with the current gating strategy, there is 54% of M1 and 82% of M2, so a total largely superior to 100%, meaning that a large majority of cells co-express both markers, and are not clearly M1 or M2. A dot plot pre-gated on CD11b+ and displaying CD86 versus CD206 should help depicting more clearly M1 and M2 macrophages on the same dot plot.

In contrast, the quantification of mRNA levels (Suppl Figure 2l) clearly shows that LPS induces CD86 expression (marker of M1), and IL4 triggers CD206 (marker of M2), as expected. In addition, S. Typhi induced CD86 (4-fold induction) but not CD206, sustaining a M1 polarization.

The authors should clarify this point, especially the gating strategy for M1/M2 and divergence between flow cytometry and mRNA analyses.

Answer: We sincerely appreciate the reviewer's thoughtful reminder. Following the comments from the reviewer, we changed the gating strategy (**Supplementary Fig. 2k**). The new gating strategy again showed that the inhibition of MCT1 and MCT4 did not affect macrophage polarization (**Figure 2m and 2n**). The updated dot plots (CD11b+CD86+CD206- and CD11b+CD206+CD86-) now clearly show the distinct M1 and M2 populations. Importantly, the overall trends remain consistent with our mRNA analyses (**Supplementary Fig. 2l**).

2/ Regarding the title:

Authors shows that transport of itaconate is facilitated by RAB32, yet this component is not specific to immune cells. The NRF2 pathway, induced by itaconate, even if it is involved in antibacterial defense mechanisms, plays also an important role in the regulation of genes that control the expression of proteins critical in the detoxication and elimination of ROS. These two pathways are not specific of immune responses. As the new data demonstrate no effect of itaconate on macrophage polarization, functionality, capacity of phagocytosis or lysosomal biogenesis (which would be pertinent immune response elements), the link between itaconate and antibacterial immune responses is really not evident and not sustain by the content of the paper. A suggestion would be: "Transport of itaconate [...] modulates macrophage' antibacterial activity".

Answer: We sincerely thank the reviewer for their valuable comments. In accordance with the suggestions, we have revised the title in the latest version of the manuscript.